# COOKBOOK: A framework for improving LLM generative abilities via programmatic data generating templates

**Avanika Narayan**[*], **Mayee F. Chen**[*], **Kush Bhatia & Christopher Ré**
Department of Computer Science
Stanford University
Stanford, CA 94305, USA
{avanikan,mfchen,kushb,chrismre}@stanford.edu

## Abstract

Fine-tuning large language models (LLMs) on instruction datasets is a common way to improve their generative capabilities. However, instruction datasets can be expensive and time-consuming to manually curate, and while LLM-generated data is less labor-intensive, it may violate user privacy agreements or terms of service of LLM providers. Therefore, we seek a way of constructing instruction datasets with samples that are not generated by humans or LLMs but still improve LLM generative capabilities. In this work, we introduce COOKBOOK, a framework that programmatically generates training data consisting of simple patterns over *random tokens*, resulting in a scalable, cost-effective approach that avoids legal and privacy issues. First, COOKBOOK uses a *template*—a data generating Python function—to produce training data that encourages the model to learn an explicit pattern-based *rule* that corresponds to a desired task. We find that fine-tuning on COOKBOOK-generated data is able to improve performance on its corresponding task by up to 52.7 accuracy points. Second, since instruction datasets improve performance on multiple downstream tasks simultaneously, COOKBOOK algorithmically learns how to mix data from various templates to optimize performance on multiple tasks. On the standard multi-task GPT4ALL evaluation suite, Mistral-7B fine-tuned using a COOKBOOK-generated dataset attains the best accuracy on average compared to other 7B parameter instruction-tuned models and is the best performing model on 3 out of 8 tasks. Finally, we analyze when and why COOKBOOK improves performance and present a metric that allows us to verify that the improvement is largely explained by the model's generations adhering better to template rules.

## 1 Introduction

Fine-tuning large language models (LLMs) on instruction datasets (Lian et al., 2023a; Longpre et al., 2023; Achiam et al., 2023; "Teknium", 2023), can significantly improve LLM generative capabilities. However, these datasets can be time-consuming and expensive to curate. Moreover, recent alternatives such as user chat logs (e.g., shareGPT) and LLM-generated data (Lian et al., 2023a; "Teknium", 2023) are less labor-intensive but can still be costly and may violate user privacy and the terms of service of LLM providers. *Programmatically generated data*, where samples are generated by programs rather than by humans or LLMs, is a potential alternative that is more scalable, cost-effective, and avoids privacy and legal issues. Examples of programmatic data include data used in Dyck languages, sequence copying, and bit parity tasks (Jelassi et al., 2024; Hahn, 2020). However, it is unclear to what extent programmatic data can be used in place of standard instruction datasets to improve model capabilities; existing work focuses on using it for understanding how models learn specific patterns and for improving pre-training, which is still followed by fine-tuning on the downstream task (Nanda et al., 2023; Wu et al., 2022). This raises the question: *is it*

---

[*]Equal contribution.

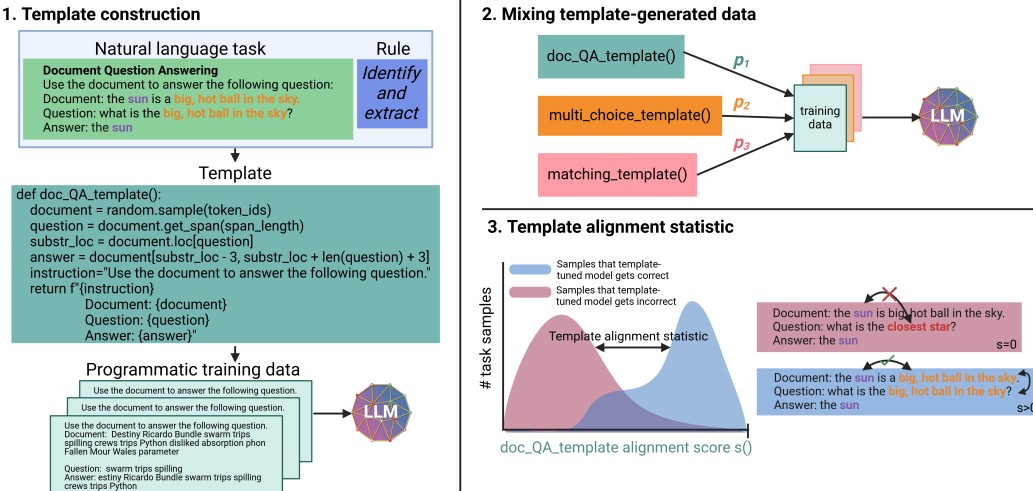

Figure 1: COOKBOOK. (1) Templates approximate a given task's "rule" and generate data consisting of patterns over random tokens. (2) Template-generated data can be mixed to improve multiple capabilities. (3) The template alignment statistic measures the extent to which the rule learned by the template is responsible for improving LLM performance.

*possible to programmatically generate an instruction dataset whose performance is competitive with human or LLM-generated instruction datasets on multiple downstream generative tasks?*

There are two key considerations for programmatically generating a performant instruction dataset.

- First, how do we design programmatic data that teaches the model rules corresponding to complex generative tasks that instruction datasets typically improve on? It is straightforward to construct programmatic data that teaches simple patterns like copying, but doing the same for more complex rules is challenging. For example, consider the document-based question answering (QA) task, where a model must answer a question about a document. One underlying rule for this task is to *identify* the relevant subpassage and *extract* an answer; it is unclear how to translate this rule into programmatic data.

- Second, since instruction datasets improve on many tasks, how do we mix programmatic data corresponding to different rules to optimize performance of multiple downstream tasks simultaneously? Recent work has shown that data mixture proportions can significantly impact downstream performance (Albalak et al., 2024). Since naive trial-and-error is costly and has a large search space, we need a principled mixing approach.

We propose COOKBOOK, a framework that explores the use of *templates*—simple Python functions—to produce programmatic training data that improves LLM performance by encouraging the model to learn explicit pattern-based rules that approximate desired generative task capabilities. Each template is designed for a given natural language (NL) task and has two key properties: 1) it invokes task-specific data generating functions, which specify how inputs and outputs are generated in order to approximate a task rule, and 2) it operates over a random token space, which not only avoids privacy and legal concerns but also allows us to express a complex rule as a pattern over the tokens. While templates are task-specific, they are composed using simple list operations—sampling, span selection, shuffling, replacement, and concatenation—and we show that their construction can be either manually specified or automatically generated using models such as GPT-4. To more clearly understand our templates, consider a rule for document QA (Figure 1) which is to identify the text in the document that pertains to the question and extract the answer from nearby. Our template for document QA constructs the document as a random sequence uniformly sampled over the token vocabulary. To instantiate the *identify and extract* rule, our template samples the question as a small span of the document token sequence and defines the answer as the span of tokens surrounding the question.

Next, to improve performance on many downstream tasks simultaneously, we discuss how to construct our instruction dataset as a mixture of samples generated from multiple templates. We propose a simple mixing algorithm over COOKBOOK templates, which we call COOKBOOK-MIX. Given COOKBOOK templates that generate data for several tasks, this algorithm first finetunes the base model on data generated by each template. Then, we use weak supervision (WS) methods (Ratner et al., 2019) to estimate the accuracy of each fine-tuned model on each downstream task without requiring labeled evaluation data. The proportions are set as the softmax of the estimated accuracies, and our instruction dataset is generated according to these proportions. Notably, these proportions are theoretically optimal under a linearity assumption on the accuracies, which we empirically validate.

Empirically, COOKBOOK programmatically generates instruction data that improves LLM performance on generative tasks, unveiling an exciting finding: *we can improve a model's natural language capabilities via standard supervised fine-tuning on patterned sequences of random tokens*. We find that Mistral fine-tuned on COOKBOOK-MIX-generated data outperforms Mistral and Llama 2 fine-tuned on existing instruction datasets (e.g., OpenOrca, OpenHermes, Capybara) on average on the multi-task GPT4ALL evaluation suite, and, on an individual task level, is the best performing model on 3 out of 8 tasks (top-1 for win-rate). We further find that fine-tuning on data generated from individual COOKBOOK templates improves performance of the corresponding NL task across 6 out of 7 downstream tasks—spanning document QA, retrieval, commonsense QA, entity matching, and entity disambiguation— on GPT-NEO-1.3B. In particular, COOKBOOK outperforms the base model by up to 52.7 points, confirming the efficacy of COOKBOOK at both the instruction dataset level and a more fine-grained single-task level.

Finally, we analyze when and why COOKBOOK can improve model performance. First, we define template alignment scorers, which quantify adherence of a NL task sample to a template rule; for instance, in document QA (Figure 1) we score samples based on how close the answer and question words are in the document (i.e., the *identify and extract* rule). Using these scorers, we develop the *template alignment statistic*, which empirically confirms that the models improvement is largely due to better adherence to template rules. Second, we study the role of the template's data generating function, random tokens, and the base model to better understand when training on COOKBOOK data generalizes to NL tasks.

## 2   Related work

We present an abbreviated related work and provide a full treatment in Appendix C. Instruction datasets range from those that are manually curated (Wang et al., 2022b; Longpre et al., 2023) to those that are generated by LLMs (Taori et al., 2023; Li et al., 2023a; Lian et al., 2023a). Programmatically generated data is an alternative to human and LLM-generated datasets, and has been explored as a replacement for manually labeled data (Ratner et al., 2020; Zhang et al., 2022a). Programmatic data is also used in pre-training, incorporating latent structures similar to those in natural language (Wu et al., 2022; Papadimitriou & Jurafsky, 2020; Krishna et al., 2021) and for understanding models (Nanda et al., 2023; Zhang et al., 2022b). For multi-task performance, recent data mixing algorithms set proportions based on online model performance, largely for pre-training (Chen et al., 2023; Xie et al., 2023).

## 3   COOKBOOK

We set up the problem of constructing a template for a single task and mixing template-generated data for multiple tasks in Section 3.1. We then describe our framework, COOK-BOOK, of task-specific templates for generating programmatic data that can improve model abilities on generative tasks (Section 3.2). In Section 3.3, we present COOKBOOK-MIX, which mixes data from several templates to improve performance on multiple tasks.

```python
def commonsense_reasoning(overlap_len: int):
    sent, c1, c2 = random.sample(token_ids)
    # inputs: use sent to get c1 and c2
    c1 = c1 + random.sample(sent, k=overlap_len)
    c2 = c2 + random.sample(token_ids, k=overlap_len)
    choices = [c1, c2].shuffle()

    # outputs: use sent and choices to get answer
    ans_idx = argmax([sent ∩ choices[0], sent ∩ choices[1]])
    answer = choices[ans_idx]

    instruction = "Select the correct choice."

    return f"""
{instruction}
{sent}
Choices:
- {choices[0]}
- {choices[1]}
Answer: {answer}"""
```

```python
def matching(noise: float):
    e1 = random.sample(token_ids)
    # inputs: use e1 to get e2
    e2 = (e1.replace(k=1) if rand.random() < 0.5
          else e1.replace(k=noise))

    # outputs: use e1 and e2 to get answer
    answer = 'yes' if e1 ∩ e2 > (1-noise) else 'no'

    instruction = "Determine whether Product A and
    Product B are the same."

    return f"""
{instruction}
Product A: {e1}
Product B: {e2}
Question: Are Product A and Product B the same?
Answer: {answer}"""
```

Figure 2: **Example templates (pseudocode)**. Templates construct the inputs, outputs, and then return a formatted sample. (Left) template for commonsense reasoning which generates two answer choices, where one choice (the answer) has a greater token overlap to the sentence. (Right) template for entity matching, which generates two entities which are labeled a match if their overlap exceeds a threshold.

## 3.1 Setup

**Constructing templates (single task).**   We are given a natural language (NL) generative task $T$ as input. Samples of $T$ have several components. Let $I \in \mathcal{I}$ be the NL instruction for the task, $x \in \mathcal{X}$ denote its inputs, and $y \in \mathcal{Y}$ denote the task output. Let $\text{fmt}_T()$ be the task formatter that formats the text sample using $x$ and $y$, prepending $I$. Using document QA as an example, $x = \{x_1, x_2\}$ where $x_1$ is the document and $x_2$ is the question, $y$ is the answer, and $\text{fmt}_T(x, y, I)$ could be f"Use the document to answer the following question. Document: {x1}. Question: {x2}. Answer: {y}". Given this information about $T$, we construct a template $G_T$, a Python function that generates samples. The goal is to design the template such that the generated samples can be used to fine-tune a model that does well on $T$. More precisely, we fine-tune a base model $f : \mathcal{X} \to \mathcal{Y}$ on $n$ samples generated from $G_T$ to yield $f_{G_T, n}$, and our goal is for $f_{G_T, n}$ to perform well on $T$.

**Mixing template-generated data (multi-task).**   We have as input $l$ templates $\boldsymbol{G_T} = \{G_{T_1}, \ldots, G_{T_l}\}$ generated using COOKBOOK for tasks $T_1, \ldots, T_l$. We use these templates to generate $n$ samples of programmatic data with mixture proportions $\boldsymbol{p} = \{p_1, \ldots, p_l\} \in \triangle^l$, the probability simplex. Define $f_{\boldsymbol{G_T}, n\boldsymbol{p}}$ as the base model $f$ fine-tuned on this mixture, with $np_i$ samples generated by each $G_{T_i}$. Suppose there are $m$ downstream evaluation tasks $\boldsymbol{T}^{\text{eval}} = \{T_1^{\text{eval}}, \ldots, T_m^{\text{eval}}\}$ (note that they may be different from the $l$ tasks that the templates were constructed for), and let $\text{acc}(f_{\boldsymbol{G_T}, n\boldsymbol{p}}, T_j^{\text{eval}})$ be the accuracy of $f_{\boldsymbol{G_T}, n\boldsymbol{p}}$ on $T_j^{\text{eval}}$. Our goal is to determine the $\boldsymbol{p}$ that maximizes average accuracy, $\underset{\boldsymbol{p} \in \triangle^l}{\text{maximize}} \frac{1}{m} \sum_{j=1}^{m} \text{acc}(f_{\boldsymbol{G_T}, n\boldsymbol{p}}, T_j^{\text{eval}})$.

## 3.2 COOKBOOK templates

We first describe the high-level process by which a template generates samples. We present examples of COOKBOOK templates in section 3.2.1, explaining how they are composed using common operators. In section 3.2.2, we explore ways to automate template creation.

**Data generation process**   The data generation process involves (1) constructing the inputs, $\hat{x}$, and (2) constructing the output $\hat{y}$ based on $\hat{x}$. Inputs are categorized into a *parent input* and *child inputs*; the parent input is a sequence of randomly sampled tokens, and the child inputs are constructed from the parent input. The output is constructed from the parent and child inputs using a data generating function that approximates the task rule. The sample is then formatted as $\text{fmt}_T(\hat{x}, \hat{y}, I)$.

### 3.2.1 Example templates

We present example templates from three generative *task families*, described below.

- **Selection** tasks involve choosing one of the given inputs (answer choices) as an output. Examples: entity disambiguation, commonsense reasoning QA, and multiple choice QA.

- **Search** tasks require extracting from the input. Examples: document QA and retrieval.

- **Comparison** tasks involve outputting "yes"/"no" based on the relationship among inputs. Example: entity matching.

For each example template, we describe its inputs (parent, child) and outputs and present corresponding pseudocode. Across the tasks, we identify five operators for generating inputs: `random.sample()`, `random.shuffle()`, `get_span()` (extract a random span from a sequence of tokens), `replace(k)` (replace tokens in a sequence with probability $k$), and list concatenation ($+$). All task templates (7 total) can be found in Appendix B.

**Document QA (Search)**   The document QA tasks involves answering a question over a provided textual context.

- **Template pseudocode**: See "doc_QA_template" in Figure 1.

- **Inputs**: document is the parent input, and `question` is the child input. document is generated as a sequence of randomly sampled tokens from the token vocabulary. `question` is generated as a subspan of document.

- **Outputs**: answer is the output which is generated by locating the `question` in the document and returning the tokens within $k$ indices of the location.

**Commonsense Reasoning (Selection)** The task of commonsense reasoning QA is to select the answer choice which best completes the sentence. For example, given a sentence of "to eat more sweets I should" and two choices of (1) "drink water" and (2) "eat more candy", the correct answer choice is "eat more candy".

- **Template pseudocode**: See "commonsense_reasoning" in left of Figure 2.

- **Inputs**: sent is the parent input, and `c1, c2` are the child inputs. sent is generated as a sequence of randomly sampled tokens from the token vocabulary. Both `c1` and `c2` are generated as sequences of randomly sampled tokens from the token vocabulary. For `c2` we append additional tokens which is a sequence of tokens sampled from sent. `choices` is the list containing `c1` and `c2` which is randomly shuffled.

- **Outputs**: answer is the index of `choices` which has the greatest overlap with sent.

**Entity Matching (Comparison)**   The task of entity matching is to determine whether two entities are equivalent.

- **Template pseudocode**: See "matching" in right of Figure 2.

- **Inputs**: The first entity `e1` is the parent input, and the second entity `e2` is the child input. `e1` is generated as a sequence of randomly sampled tokens from the token vocabulary. With 50% probability, `e2` is set to `e1` with a small amount of tokens replaced (as determined by the noise threshold ); otherwise, `e2` is generated as a sequence of randomly sampled tokens from the token vocabulary of the same length as `e1`.

- **Outputs**: answer is determined by the amount of overlap between `e1` and `e2`. If the amount of overlap is above a threshold, the output is "yes". If not, it is "no".

Note that across all templates, the task rules are approximated by translating them into patterns that involve token overlap; that is, outputs are constructed from inputs based on token overlap among the inputs. However, we do not claim that the token overlap and the five operators above are necessary components of a template; in section B.7 we present a poetry generation task and template, which does not invoke most of these components.

### 3.2.2   Automating template creation

We introduce a method for automatically generating COOKBOOK templates using GPT-4. We do so by prompting the model with a description of the data generation process (see Section 3) alongside two in-context examples mapping a task description to a data generating template (see Appendix E for prompt details). We evaluate the performance of GPT-NEO-1.3B finetuned on data generated from these templates — which we call

COOKBOOK-NEO-AUTO — finding that performance of COOKBOOK-NEO-AUTO is within 0.2% of manually curated templates (see Appendix E for details).

### 3.3 COOKBOOK-MIX: mixing template-generated data

Given multiple templates and downstream tasks, we study how to set template proportions to construct a training dataset that improves model performance across all the downstream tasks. First, we impose a simple linearity assumption on model accuracy and derive the optimal data proportions $p^\star$ that maximize average downstream accuracy of the COOKBOOK-tuned model. This approach requires evaluating models on labeled data, so we present an extension where we can approximate $p^\star$ using unlabeled data via a latent variable model inspired by weak supervision (Ratner et al., 2019). Our full algorithm, COOKBOOK-MIX, is provided as Algorithm 1 in Appendix D.1.

**Optimal template-generated data proportions.** Our objective is to maximize $f_{G_T, np}$'s average downstream accuracy, $\underset{p \in \triangle^l}{\text{maximize}} \frac{1}{m} \sum_{j=1}^{m} \text{acc}(f_{G_T, np}, T_j^{\text{eval}})$. To solve this problem, we impose a simple linear assumption: that $\text{acc}(f_{G_T, np}, T_j^{\text{eval}}) = \sum_{i=1}^{m} p_i \text{acc}(f_{G_{T_i}, n}, T_j^{\text{eval}})$ for all $j \in [m]$. That is, the accuracy of a model trained on a weighted mixture is equal to the weighted average of models trained on each template, which we empirically assess in Appendix D.2. We also add an entropy term $H(p) = -\sum_{i=1}^{l} p_i \log p_i$ with a weight $\eta \geq 0$ to control how close to uniform $p$ should be. Our optimization problem is now

$$\underset{p \in \triangle^l}{\text{maximize}} \frac{1}{m} \sum_{j=1}^{m} \sum_{i=1}^{l} p_i \text{acc}(f_{G_{T_i}, n}, T_j^{\text{eval}}) + \eta H(p). \tag{1}$$

Solving this optimization problem yields the following solution.

**Proposition 1.** *Define $A \in \mathbb{R}^{l \times m}$ where $A_{ij} = \text{acc}(f_{G_{T_i}, n}, T_j^{\text{eval}})$. Let $\sigma_i = \exp(\frac{1}{m\eta} \sum_{j=1}^{m} A_{ij})$ for all $i \in [l]$. Then, the $p^\star$ that maximizes (1) satisfies $p_i^\star = \frac{\sigma_i}{\sum_{k=1}^{l} \sigma_k}$ for all $i \in [l]$.*

See Appendix D.3 for the proof. The computation of $p^\star$ is straightforward. We train one model per template, $f_{G_{T_i}, n}$ for each $i \in [l]$. We compute the average accuracy across the $m$ downstream tasks per model, and then compute the softmax over these accuracies to get the proportions $p^\star$ to determine how many samples are needed from each template.

**Extension to evaluation data without ground-truth outputs.** Note that computing $\text{acc}(f_{G_{T_i}, n}, T_j^{\text{eval}})$ requires evaluating on a dataset with ground-truth outputs. However, in practice datasets are not often annotated with outputs, which requires us to estimate the accuracy of the model. Since we have multiple individual models, we can frame them as noisy "voters" that make predictions on the true label and cast accuracy estimation as a weak supervision problem, a line of work that estimates labels given an unlabeled dataset and several heuristic voters using latent variable models. We describe this extension of our approach, which uses the MeTaL weak supervision algorithm (Ratner et al., 2019) in Appendix D.1.

### 3.4 Training on COOKBOOK-data

All models trained on COOKBOOK-template data were finetuned by running a hyper-parameter search, sweeping across learning rate ($[4e - 06, 5e - 06, 8e - 06, 5e - 05, 8e - 05]$), batch size ($[8, 16, 32, 64]$) and total training steps ($[100, 200, 300, 400 \text{ and } 500]$). In finetuning COOKBOOK-models, no training samples were repeated.

## 4 Experimental evaluations

We evaluate the empirical performance of COOKBOOK in a multi-task evaluation setting (i.e., against standard instruction datasets on many downstream tasks) and in a single-task evaluation setting on NL tasks that our COOKBOOK templates correspond to. In Appendix F, we show that COOKBOOK can be extended to creative tasks (i.e., poetry generation) going beyond select, search, and compare tasks.

| Model | arc_c | arc_e | boolq | hellaswag | lambada | openbookqa | piqa | winogrande | average |
|---|---|---|---|---|---|---|---|---|---|
| LLAMA-2-7B | 46.25 | 74.58 | 77.74 | 75.99 | 73.92 | 44.20 | 79.11 | 69.14 | 67.61 |
| LLAMA-2-7B-NH | 49.74 | 76.09 | 80.00 | 77.72 | 72.99 | **46.40** | 79.76 | 70.01 | 69.09 |
| MISTRAL-7B | 54.10 | 79.50 | 83.49 | 81.12 | 75.59 | 44.40 | 82.05 | 73.88 | 71.76 |
| MISTRAL-7B-ORCA | 56.14 | 79.59 | 86.57 | 81.73 | 72.37 | 45.60 | **83.03** | 73.24 | 72.28 |
| MISTRAL-7B-OH | **59.98** | 81.65 | **86.73** | **81.77** | 73.90 | 44.20 | 82.70 | 73.56 | 73.06 |
| COOKBOOK-LLAMA | 48.04 | 76.77 | 79.20 | 76.04 | 77.10 | 43.40 | 78.56 | 69.30 | 68.55 |
| **COOKBOOK-MIST** | 57.76 | **83.21** | 85.23 | 80.99 | **78.23** | 44.00 | 82.32 | **74.27** | **73.25** |

Table 1: **Performance on GPT4ALL benchmark.** We denote our COOKBOOK-MIX-tuned MISTRAL-7B model as COOKBOOK-MIST. Averaged across tasks, COOKBOOK-MIST has the best accuracy. For baseline datasets, "NH" denotes NousHermes, "OH" is Open-Hermes, and "ORCA" is OpenOrca.

## 4.1 Multi-task Evaluation

We evaluate if the COOKBOOK framework can improve multiple generative capabilities at once. Here, we compare COOKBOOK-MIX to SoTA instruction datasets (e.g., OpenOrca, OpenHermes).

**Our method.** We fine-tune two pre-trained base models, LLAMA-2-7B (Touvron et al., 2023) and MISTRAL-7B (Jiang et al., 2023). The set of templates we consider is: MATCHING, ENTITY-DISAMBIGUATION, MULTI-CHOICE-QA, and COMMONSENSE-SELECT (refer to Appendix B for their constructions). We fine-tune each base model on these templates using the data proportions obtained using COOKBOOK-MIX as described in Section 3.3.

**Evaluation datasets.** We evaluate on the standard GPT4ALL (NomicAI) benchmark considered by many models (e.g., Mamba (Gu & Dao, 2023), Cerebras-GPT (Dey et al., 2023)). We report average zero-shot accuracy for all tasks using the standard EleutherAI LM Evaluation Harness library (LM eval) (Gao et al., 2023). The LM eval harness provides a fixed set of prompt formats for all tasks, generating accuracy and standard deviation metrics.

**Baselines.** As baselines we consider the base LLAMA-2-7B and MISTRAL-7B models. Additionally, we compare against the current SoTA open-source instruction-tuned versions of these models (OpenHermes-Mistral-7B (Teknium, 2023), Mistral-7B-OpenOrca (Lian et al., 2023b)) and the closed source versions (Nous-Llama-2-7b (Research, 2023b), Nous-Capybara-7B (Research, 2023a)).

**Results.** Results for our multi-task evaluations can be found in Table 1, with full results in Table 5. Averaging across tasks we find that (1) COOKBOOK-MIX improves performance across model variants—Llama and Mistral (see Table 1), (2) MISTRAL-7B fine-tuned with COOKBOOK-MIX (which we abbreviate COOKBOOK-MISTRAL) is the best fine-tuned model on the benchmark suite and also outperforms other mixtures of COOKBOOK templates, such as a uniform mixture (Table 6), and (3) COOKBOOK-MISTRAL is the best performing model on 3/8 tasks.

## 4.2 Single-task Evaluation

This section evaluates single-task performance of task-specific COOKBOOK templates, offering a more fine-grained study of the templates themselves without mixing.

| Dataset | NEO-BASE | NEO-FEW | COOKBOOK-NEO |
|---|---|---|---|
| TYDIQA | $21.8 _{\pm 0.49}$ | $14.0 _{\pm 2.4}$ | $\textbf{41.9} _{\pm 0.40}$ |
| SQUAD | $14.5 _{\pm 0.78}$ | $50.4 _{\pm 5.2}$ | $\textbf{60.5} _{\pm 0.68}$ |
| PIQA | $0.0 _{\pm 0.0}$ | $48.5 _{\pm 0.5}$ | $\textbf{52.7} _{\pm 0.43}$ |
| MS_MARCO | $12.6 _{\pm 1.08}$ | $17.7 _{\pm 1.0}$ | $\textbf{18.6} _{\pm 0.16}$ |
| WINOGRANDE | $3.9 _{\pm 0.25}$ | $64.6 _{\pm 32.4}$ | $54.3 _{\pm 0.87}$ |
| BEER | $26.7 _{\pm 0.0}$ | $26.7 _{\pm 0.0}$ | $\textbf{66.6} _{\pm 0.0}$ |
| ITUNES-AMAZON | $39.7 _{\pm 0.0}$ | $63.1 _{\pm 0.0}$ | $\textbf{69.6} _{\pm 0.0}$ |

Table 2: **Single-task evaluations**. We report accuracy for all datasets except BEER and ITUNES-AMAZON (F1-score). COOKBOOK-tuned models exhibit improved performance over the base model.

**Our method.** For each evaluation task, we generate data from the associated COOKBOOK template (see Table 8 for the mapping between task and template). We fine-tune two base pre-trained models, GPT-NEO-1.3B (Gao et al., 2020) and CEREBRAS-GPT-1.3B (Dey et al., 2023), using data generated from the template. In our data generating process, a new batch is sampled at each step. Across tasks, we fine-tune models over an average of 4.2K datapoints. We evaluate the fine-tuned model's zero-shot performance.

**Evaluation datasets.** For our evaluations, we consider 7 tasks across selection (PIQA, WINOGRANDE), search (TYDIQA, SQUAD, MS_MARCO) and comparison (BEER, ITUNES-AMAZON) categories. For all tasks excluding BEER and ITUNES-AMAZON, for which we report F1-score, we report accuracy. Table 7 in Appendix G.5.2 contains dataset statistics.

**Baselines.** We compare our COOKBOOK-tuned models to the base model via two baselines: zero-shot and few-shot (where $k$=3) using natural language task samples as in-context examples. Zero-shot involves directly evaluating the base model's ability on a given task, while few-shot primes the base model with task-specific formatting and context.

**Results.** Table 2 shows the results for GPT-NEO-1.3B. We defer results for CEREBRAS-GPT-1.3B to Table 9 in Appendix G.5.2. Our results show that (1) COOKBOOK-NEO outperforms the zero-shot and ICL performance on 6/7 tasks, (2) performance of COOKBOOK models can be 52.7 points better than base zero-shot performance, suggesting that COOKBOOK-generated data can help the model learn a task it was previously unable to do, (3) similar trends can be seen across model families (see Appendix G.5.2).

## 5 Analysis of COOKBOOK

First, we propose and measure the template alignment statistic on several tasks to validate that the rules taught by templates are responsible for improved task performance. Then, we analyze the key components of COOKBOOK to better understand its effectiveness, finding that 1) the extent of NL knowledge the base model has impacts COOKBOOK's performance; 2) training on data from one COOKBOOK template can improve on multiple tasks; 3) rules, not only the instruction formatting of samples, are necessary to the performance of COOKBOOK.

### 5.1 Template Alignment Framework

We introduce the template alignment scorer, which measures how similar a NL sample and a template are. We use this to define the template alignment statistic, which measures how better adherence to the template's rule is responsible for improved performance on the task.

**Template Alignment Scorer** Given a task $T$ and its corresponding template $G_T$ as constructed in Section 3, we propose a template alignment scorer $s_{G_T} : \mathcal{X} \times \mathcal{Y} \to [0, 1]$. The template-specific scorer captures how well the input sample's relationship between $x$ and $y$ follows that of $G_T$'s data generating functions. We provide three examples below:

- **Document QA**: the template selects a random span of a document as the question and defines the answer as the surrounding span. $s_{G_T}$ scores a sample by finding where the answer is located in the document and computing the fraction of words in the question that are near the answer (Fig 1). Note that template-generated samples have a score of 1.

- **Commonsense Reasoning**: the template generates two answer choices, with only one choice containing tokens from the sentence. $s_{G_T}$ scores a sample by finding the words that are unique to the two choices, measuring the minimum normalized embedding distance between each choice's unique words and the words in the sentence, and computing the absolute value of the difference between the two choices' distances to the input sentence. This quantity is large if one answer choice has much higher word similarity to the input sentence than the other. Note that when we use the hamming distance, template-generated samples have a score of 1.

- **Entity Matching**: the template first generates one entity randomly, and then generates the second entity to have some overlap with the first one. $s_{G_T}$ scores a sample by computing the word overlap between the two entities. Note that template-generated samples have a score equal to $1 - \texttt{noise}$ when the entities are equivalent.

**Template Alignment Statistic**   Next, we use the alignment scorer to relate model performance to template rule adherence. Recall the base model $f : \mathcal{X} \rightarrow \mathcal{Y}$ and the COOK-BOOK-tuned model $f_{G_T,n}$. Given NL task samples $\mathcal{D}_T$, let $\mathcal{D}_T^+(G_T) = \{(x, y) \in \mathcal{D}_T : f(x) \neq y, f_{G_T,n}(x) = y\}$ be the set of samples that $f$ gets incorrect and $f_{G_T,n}$ gets correct, and let $\mathcal{D}_T^-(G_T) = \{(x, y) \in \mathcal{D}_T : f(x) \neq y, f_{G_T,n}(x) \neq y\}$ be the set of samples that both $f$ and $f_{G_T,n}$ get incorrect. We now define the template alignment statistic, which characterizes how different the template alignment scores are on samples that the COOKBOOK-tuned model gets correct versus incorrect.

**Definition 1.** *Let* $S_T^+(G_T) = \{s_{G_T}(x, y) \quad \forall (x, y) \in \mathcal{D}_T^+(G_T)\}$ *and* $S_T^-(G_T) = \{s_{G_T}(x, y) \ \forall (x, y) \in \mathcal{D}_T^-(G_T)\}$ *be the alignment scores over* $\mathcal{D}_T^+(G_T)$ *and* $\mathcal{D}_T^-(G_T)$, *respectively. Define* $F_{T,G_T}^+ : [0, 1] \rightarrow [0, 1]$ *as the empirical CDF over* $S_T^+(G_T)$ *and similarly define* $F_{T,G_T}^-$. *Then, the template alignment statistic for task T, template* $G_T$ *is*

$$\mathcal{A}(T, G_T) = \sup_{s \in [0,1]} |F_{T,G_T}^+(s) - F_{T,G_T}^-(s)|. \tag{2}$$

Note that $\mathcal{A}(T, G_T)$ is the total variation distance between the empirical distributions of the template alignment scores for $\mathcal{D}_T^+(G_T)$ and $\mathcal{D}_T^-(G_T)$, as well as the Kolmogorov-Smirnov test statistic for the two-sample hypothesis test on whether the scores for these two subsets of $\mathcal{D}_T$ come from the same distribution. This statistic is thus bounded between 0 and 1. A large value for this statistic suggests that there is significant difference between improved and non-improved samples in terms of how they adhere to template rules.

| Template $G_T$ / Task $T$ | TEMPLATE-MISTRAL $\mathcal{A}(T, G_T)$ |
|---|---|
| **PIQA / COMMONSENSE-SELECT** | $(0.867, 7.1 \times 10^{-46})$ |
| **DBLP-ACM / MATCHING** | $(1.0, 0.0025)$ ) |
| **TYDIQA / DOCUMENT-QA** | $(0.615, 0.013)$ |

Table 3: **Template alignment statistics.** Template alignment statistics for COOKBOOK-tuned models. Values are the (template alignment statistic, p-value). COOKBOOK-tuned models have learned the rule captured by the synthetic.

**Results.**   We compute the template alignment statistic for the MISTRAL-7B-template models on the tasks TYDIQA, PIQA and DBLP-ACM using the DOCUMENT-QA, COMMONSENSE-SELECT and MATCHING templates respectively. Note that for PIQA, we use Sentence-BERT embeddings (Reimers & Gurevych, 2019) to compute embedding distance between the words in the answer choices and the words in the goal. We find that the template alignment statistic for each task (Table 3) is statistically significant ($< 0.05$), suggesting that rules taught by the templates are responsible for improvement on downstream NL tasks.

## 5.2 Understanding COOKBOOK random tokens and rules

**When do random tokens work?** We hypothesize that fine-tuning on random tokens helps learn a NL task when the model already has sufficient NL capabilities. To test this, we evaluate COOKBOOK on PYTHIA-1B checkpoints (log scale from 0 to 144K) as the base models, where later model checkpoints have more sufficient NL capabilities. We do this on PIQA, ITUNES-AMAZON, SQUAD, WINOGRANDE, and their respective templates, finding that there exists a checkpoint at which COOKBOOK starts to help performance significantly, before which it has little effect (see Figure 4 in Appendix H). This suggests that the generalization from random tokens to NL is dependent on the base model's NL capabilities.

**Do rules taught over random tokens result in less overfitting?** We compare the *overfitting* of rules taught over random tokens versus rules taught over NL tokens. We fine-tune models on the COOKBOOK MATCHING template and on the NL matching task itself (ITUNES-AMAZON), and evaluate them on ITUNES-AMAZON and two other tasks, PIQA and TYDIQA (see Table 11). We observe that COOKBOOK-tuning actually improves base model performance across tasks beyond matching (up to 5 points), whereas NL-tuning worsens base performance (by up to 7 points) for all tasks outside of matching. This suggests that skills

taught in the random token space help generalize to multiple tasks and overfit less, offering a potential explanation for how COOKBOOK-MIX outperforms standard instruction datasets.

**Do we need data generating functions: Are random tokens all we need?** We empirically inspect the importance of the template rules vs. the task format ($\text{fmt}_T$) in improving downstream performance by replacing our rule-based generated input and outputs in $\text{fmt}_T()$ with random tokens without any structured patterns. Our results show that finetuning on data without rules leads to worsened performance — an average performance drop of 18.3 accuracy points (see Table 12 in Appendix H).

## 6 Discussion

In this work, we tackle the challenge of building instruction datasets for improving generative abilities. Via COOKBOOK, we show that it is possible to programmatically generate data that teaches task-specific rules. Moreover, we show how such programmatic data can be mixed to improve performance on many tasks at once. Finally, we propose a method for measuring whether a model has indeed learned a task rule, and whether learning that rule improves downstream task performance. In future work, we seek to further explore how to use programmatic data generation to enable self-improving systems.

## 7 Acknowledgements

We thank Michael Zhang, Neel Guha, Michael Wornow, Ben Spector, Silas Alberti, Jordan Juravsky, Eric Nguyen, and Alyssa Unell for their feedback and discussion.

We gratefully acknowledge the support of NIH under No. U54EB020405 (Mobilize), NSF under Nos. CCF2247015 (Hardware-Aware), CCF1763315 (Beyond Sparsity), CCF1563078 (Volume to Velocity), and 1937301 (RTML); US DEVCOM ARL under Nos. W911NF-23-2-0184 (Long-context) and W911NF-21-2-0251 (Interactive Human-AI Teaming); ONR under Nos. N000142312633 (Deep Signal Processing); Stanford HAI under No. 247183; NXP, Xilinx, LETI-CEA, Intel, IBM, Microsoft, NEC, Toshiba, TSMC, ARM, Hitachi, BASF, Accenture, Ericsson, Qualcomm, Analog Devices, Google Cloud, Salesforce, Total, the HAI-GCP Cloud Credits for Research program, the Stanford Data Science Initiative (SDSI), Knight-Hennessy Scholarship, NSF Graduate Research Fellowship and members of the Stanford DAWN project: Meta, Google, and VMWare. The U.S. Government is authorized to reproduce and distribute reprints for Governmental purposes notwithstanding any copyright notation thereon. Any opinions, findings, and conclusions or recommendations expressed in this material are those of the authors and do not necessarily reflect the views, policies, or endorsements, either expressed or implied, of NIH, ONR, or the U.S. Government.

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

## Appendix

We provide an overview of the sections of the Appendix:

- Appendix A: we provide a glossary of the variables and symbols used in this paper.
- Appendix B: we present templates corresponding to generative tasks as well as a sample data point generated from each template.
- Appendix C: we provide a full related work.
- Appendix D: we provide more details on COOKBOOK-MIX, including the formal algorithm, experiments to verify the linearity assumption, and a proof of Proposition 1.
- Appendix E: we explain how to use GPT-4 to automatically generate COOKBOOK templates and evaluate these generated templates.
- Appendix F: we demonstrate how the COOKBOOK framework can be extended to a more creative poetry generation task.
- Appendix G: we provide additional details on the experiments in Section 4.
- Appendix H: we provide additional experiments and details for understanding COOKBOOK.

## A  Notation

The glossary is given in Table 4 below.

| Symbol | Used for |
|---|---|
| $T$ | Natural language (NL) task that we aim to construct a template for improving. |
| $I$ | Natural language instruction $I \in \mathcal{I}$ for task $T$. |
| $x$ | Inputs $x \in \mathcal{X}$ for task $T$. |
| $y$ | Output $y \in \mathcal{Y}$ for task $T$. |
| $\mathrm{fmt}()$ | Function that formats instruction, inputs and output for task $T$ into $\mathrm{fmt}(x, y, I)$. |
| $G_T$ | Data generating template corresponding to task $T$. |
| $f$ | Base model $f : \mathcal{X} \to \mathcal{Y}$ whose generative capabilities we aim to improve. |
| $n$ | Number of samples generated from templates used for fine-tuning $f$. |
| $f_{G_T,n}$ | Base model $f$ fine-tuned on $n$ samples generated by template $G_T$. |
| $\boldsymbol{G_T}$ | A set of $l$ templates $\boldsymbol{G_T} = \{G_{T_1}, \ldots, G_{T_l}\}$ used as input for the data mixing multi-task evaluation setting. |
| $\boldsymbol{p}$ | Mixture proportions $\boldsymbol{p} = \{p_1, \ldots, p_l\} \in \triangle^l$ over data generated from $\boldsymbol{G_T}$. |
| $f_{\boldsymbol{G_T},n\boldsymbol{p}}$ | Base model $f$ fine-tuned on $n$ samples, with $np_i$ samples from each $G_{T_i}$. |
| $\boldsymbol{T}^{\mathrm{eval}}$ | A set of $m$ downstream tasks $\boldsymbol{T}^{\mathrm{eval}} = \{T_1^{\mathrm{eval}}, \ldots, T_m^{\mathrm{eval}}\}$ we aim to improve performance on. |
| $\mathrm{acc}(f_{\boldsymbol{G_T},n\boldsymbol{p}}, T_j^{\mathrm{eval}})$ | The accuracy of $f_{\boldsymbol{G_T},n\boldsymbol{p}}$ on $T_j^{\mathrm{eval}}$. |
| $\eta$ | Weight $\eta \geq 0$ on entropy term $H(\boldsymbol{p})$ in (1) controlling how close to uniform $\boldsymbol{p}$ should be. |
| $s_{G_T}$ | Template alignment scorer $s_{G_T} : \mathcal{X} \times \mathcal{Y} \to [0, 1]$ that measures how much a NL sample from $T$ adheres to the rule of template $G_T$. |
| $\mathcal{A}(T, G_T)$ | Template alignment statistic for task $T$, template $G_T$ which measures the total variation distance in template alignment scores for samples of $T$ that the model fine-tuned on data from $G_T$ gets correct versus incorrect (see def. 1). |

Table 4: Glossary of variables and symbols used in this paper.

# B Templates

We present seven templates (MATCHING, MULTI-CHOICE-QA, DOCUMENT-QA, ENTITY-DISAMBIGUATION, COMMONSENSE-SELECT, TOKEN-RETRIEVAL, and POETRY GENERATION) and a sample data point generated from each.

## B.1 Matching

**Data generating template:**

```
def matching(noise: int):
    import random

    e1 = random.sample(token_ids, k=1)

    # inputs: use e1 to get e2
    e2 = e1.replace(k=1) if random.random() < 0.5 else e1.replace(k=noise
    ↪ )

    # outputs: use e1 and e2 to get output
    answer = 'yes' if len(set(e1) & set(e2)) > (1 - noise) * len(e1) else
    ↪ 'no'
    instruction = "Determine whether product A and product B are the same
    ↪ .\n"

    return f"""
        {instruction}
        Product A: {e1}
        Product B: {e2}
        Question: Are Product A and Product B the same?
        Answer: {answer}"""
```

**Sample data point:**

```
Determine whether product A and product B are the same
Product A:  occur Competing TObuiltembean Hollywood met
Product B:  occur Competing TObuiltembean Hollywood met
Question: Are Product A and Product B the same?
Answer: yes
```

## B.2 Multi-Choice QA

**Data generating template:**

```
def multi_choice_qa (overlap_len: int):
  question = random.sample(token_ids)
    (c1, c2, c3, c4, c5) = random.sample(token_ids, k=5)

  # inputs: use question to get correct choice c5
    c5 = question.sample(k=overlap_len) + c5[:-overlap_len]

  # outputs: use  question, [c1, . . ., c5]) to get the answer
    choices = [c1, c2, c3, c4, c5].shuffle()
  ans_idx = argmax([question \cap choices[0],..., question \cap choices
    ↪ [4]])
  answer = choices[ans_idx]

  instruction = "Answer the question.\n"
  return f"""
  {instruction}
  Question: {question}
```

```
Choices:
- {choices[0]}
- {choices[1]}
- {choices[2]}
- {choices[3]}
- {choices[4]}
Answer: {answer}"""
```

**Sample data point**:

```
Answer the question.
Why lake Shares wildly Gandhi rers ademic AES 1995 ports?
Choices:
- poke installment
- ORS> unmanned Slave ellar Mart English
- visions AES wildly lakeexper Gamer Gate ports ademicE
- Haibeh Dom
- aiming 462 adultery Greenberg collar
Answer:  visions AES wildly lakeexper Gamer Gate ports ademicE
```

## B.3  Document QA

**Data generating template:**

```
def qa_template(min_slen: int, max_slen: int):
  document = random.sample(token_ids) #input source

  # inputs: use document to get question
  span_length = random.choice(min_slen, max_slen)
  question = document.get_span(k=span_length) # get indexes

  # outputs: use document and question to get the answer
  answer = document[loc(document.intersect(question)) - 3 : loc(document.
    ↪ intersect(question)) + 3]

    instruction = "Use the document to answer the question.\n"

    return f"""{instruction}
  Document: {document}

  Question: {question}
Answer: {answer}""
```

**Sample data point**:

```
Use the document to answer the following question.
Document:  Destiny Ricardo Bundle swarm trips spilling crews
trips Python disliked absorption phon Fallen Mour Wales
parameter

Question:  swarm trips spilling
Answer: estiny Ricardo Bundle swarm trips spilling crews trips
Python
```

## B.4  Entity Disambiguation

**Data generating template:**

```python
def entity_disambiguation(e_span_len: int):
  sentence_1, sentence_2 = random.sample(token_ids)

  # inputs: use sentence_1 to set sentence_2, e1 and e2
  s1, s2 = sentence_1.get_span(k=e_span_len)
  e1, e2 = s1[0], s2[0]

  if random.rand() < 0.5
    sentence_2 +=['<blank>'] + s1[1:])
  else:
    sentence_2 +=['<blank>'] + s2[1:])

  # outputs: use (question, sentence_1, sentence_2) to get the answer
  support = sentence_2.split('<blank>')[-1]
  span_loc = loc(sentence_1.intersect(support) - 1]
  answer = sentence_1[span_loc - 1]

  instruction = "Select the choice which best completes the <BLANK>.\n"
  return f"""{instruction}
{sentence_1}.{sentence_2}
Choices:
- {e1}
- {e2}
Answer: {answer}"""
```

**Sample data point**:

```
Select the phrase which best fills in the <BLANK>.
Sentence:  ranc Islands solid illustrates horsepower furry Pay
early Santiago *)scan output. ographies deals privatization
<BLANK>  Santiago *)scan output.
Choices:
-  Islands
-  early
Answer:  early
```

## B.5 Commonsense Selection

**Data generating template:**

```python
def commonsense_qa(overlap_len: int):
  question = random.sample(token_ids)
  c1 = random.sample(token_ids)
    c2 = random.sample(token_ids)

  # inputs: use question to set c1 and c2
  c1 = c1 + question.sample(k=overlap_len)
  c2 = c2 + random.sample(token_ids, k=overlap_len)
  choices = [c1, c2].shuffle()

  # outputs: use (question, [c1, . . ., c5]) to get the answer
  ans_idx = argmax([len(question.intersect(choices[0])), len(question.
    ↪ intersect(choices[1]]))
  answer = choices[ans_idx]

  instruction = "Answer the question.\n"
  return f"""
{instruction}
Question: {question}
```

```
Choices:
- {choices[0]}
- {choices[1]}
Answer: {answer}"""
```

**Sample datapoint:**

```
Select the choice which best completes the sentence.
midst Gloss Quant Nest engaging Soul Customer
Choices:
- Zack extraordinarily willingly Gene Quant engaging rest
- Zack extraordinarily willingly Gene With No DXwaters shone
Answer: Zack extraordinarily willingly Gene Quant engaging rest
```

### B.6  Token Retrieval

**Data generating template:**

```
def token_retrieval(overlap_len: int):
  docs = []
  for i in range(10):
    docs.append(random.sample(token_id)

  # inputs: use docs to set question
  idx = random.randint(0,10) # randomly sample an index from 0...9
  doc_with_answer = docs[idx]
  question = doc_with_answer.sample(overlap_len)

  #outputs: use (question, docs) to get the answer
  answer_idx = argmax[question.intersect(docs[0]),...,question.intersect(
    ↪ docs[9])
  answer = docs[answer_idx]

  instruction = "Use the documents to answer the question.\n"
  return f"""
{instruction}
{docs[0]}
{docs[1]}
...
{docs[9]}
Question: {question}
Answer: {answer}"""
```

**Sample datapoint:**

```
Use the documents to answer the question.
Document 0: Def culture Luc writers takingo  adjustment
Document 1: saturally bites song house speak
Document 2: BEATE level arbitrator Layer eminent substant
Document 3: spend conesz identification unincorporateddemand
Document 4: incident existent Back Fellow Turk peacea Father
Document 5: citations Cove solel Nick cards removing comprise
Document 6: quilt sun instructed facil enacted council confess
Document 7: cs probably 59 specify Page harris famine pumps
Document 8: equal Originally quick Adjust hearted OCK beat
Document 9: Base Sept 168 '</resh593Mr tang moss mash pain

Question: 593Mr tang moss mash pain Dir
Answer: Base Sept 168 '</resh593Mr tang moss mash pain
```

### B.7 Poetry Generation

**??) Data generating template:**

```python
def rhyme_scheme(rhyme_dict: dict):
  # inputs
  topic = random.sample(token_ids, k=1)
  rhyme_scheme_A, rhyme_scheme_B = rhyme_dict.sample(k=2)

  # inputs: use (topic, rhyme_scheme_A, rhyme_scheme_B) to get lines
  lines = []
  for idx in range(5):
    r_word = rhyme_scheme_A.sample() if idx % 2 == 0 else rhyme_scheme_B.
    ↪ sample()

    line = random.sample(token_ids)
    line = line.insert(topic) + r_word
        lines.append(line)

  instruction = "Write a five line poem with an ABABA rhyme scheme about"

    return f"""
      {instruction} {topic}
      {lines[0]}
      {lines[1]}
      ...
      {lines[4]}"""
```

**Sample datapoint:**

```
Write a five line poem with a ABABA rhyme structure about abi
abi abilities Ches 76 Purabul befriend
abi agre Chung Rapt unfit hate redditt
item Struabi pre transcend
Pedro Rescueabi 1080 vines296speed ledet
continents anticip EMP texts sigabi trend
```

## C  Complete Related Works

There is a large body of work which studies instruction-tuning dataset creation; programmatic data for approximating rules, pre-training, and understanding; and data mixing. Here we review some of them which relate most closely with our work.

**Instruction tuning** Instruction tuning datasets seek to improve the ability of LLMs to follow instructions (i.e., "Generate a summary of the following news article"). These datasets consist of input and output pairs, where the input is composed of an instruction with some context and the output is the "gold" generation. Early instruction tuning datasets utilized manual data curation to construct these <instruction, input, output> triples for tasks such as summarization (Bai et al., 2022). In an effort to scale the instruction tuning dataset curation process, new datasets emerged which prepend natural language instructions to standard NLP tasks where the input-output pairs previously exist, such as Supernatural Instructions (Mishra et al., 2022; Wang et al., 2022b), the Flan Collection (Longpre et al., 2023), and in Chung et al. (2022). More recently, instruction tuning datasets have emerged which use existing LLMs (e.g., GPT-4 Achiam et al. (2023)) to generate the data, such as OpenOrca, Alpaca, Wizardcoder, Camel, Amplify-instruct, Self-instruct, and GLAN (Lian et al., 2023a; Taori et al., 2023; Luo et al., 2023; Li et al., 2023a; Daniele & Suphavadeeprasit, 2023; Wang et al., 2022a; Li et al., 2024). In this approach, instructions are fed to the LLM (i.e., "Generate a list of 5 animals") and the model generates an output. While this approach sidesteps manual data curation efforts, it suffer from privacy and terms of service violations. COOKBOOK circumvents the need for both manual or model-generated data. Some recent

approaches also train their own models to generate instruction data, such as instruction backtranslation and Bonito (Li et al., 2023b; Nayak et al., 2024). While these can avoid legal issues, it can still be computationally expensive to use an LLM to generate a large corpus of instruction data. In contrast, COOKBOOK's templates generate data without the inference costs of using an LLM.

**Programmatic data for approximating rules**: Programmatic data generation has been primarily studied for the automated *labeling of datasets* (Ratner et al., 2020; 2016; Hearst, 1992), where heuristics are used to approximate the true labeling rule. For example, a heuristic for Youtube spam comment classification is to check if the word "subscribe" is in the comment (Zhang et al., 2021). These works provide programmatic labels for classification tasks given an unlabeled dataset containing inputs only, and they find that training on data with these programmatic labels can produce models that do well on the given task. However, these approaches do not directly extend to our setting, which requires us to construct entire samples for generative tasks—both programmatic inputs and outputs, which are often open-ended generations (e.g. document QA).

**Programmatic data for improved training** Prior work has shown the value of synthetic, token-level tasks such as set operations as an alternate to natural language data for LLM pre-training (Wu et al., 2022). Similarly, other work has demonstrated the efficacy of data with latent structure (e.g., music) in the pretraining phrase (Papadimitriou & Jurafsky, 2020) as well as for tasks such as summarization (Krishna et al., 2021). These works are similar to COOKBOOK in that they utilize non-language based inputs to improve LLM performance, but unlike COOKBOOK, which directly evaluates the model trained on programmatic data, they all require some fine-tuning on natural language data afterwards. Additionally, programmatic data has be shown to improve classification abilities by using it to train a synthetic reasoning module and compose it with a base LLM (Bhatia et al., 2023); however, this approach does not apply to generative tasks.

**Programmatic data for LLM understanding** Recent works have explore the use of synthetic tasks for understanding the underlying mechanisms of LLMs (Bricken et al., 2023; Nanda et al., 2022; Ravfogel et al., 2019; Zhang et al., 2022b) as well as for understanding and improving architectures (Fu et al., 2022; Arora et al., 2023; Gu & Dao, 2023; Poli et al., 2023). Unlike these prior works which seek to use non-language based tasks to understand LLMs, we use such data to improve LLM capabilities.

**Data mixing** An important step in curating training data is to determine the mixture over several given groups of data (e.g., domains such as CommonCrawl, ArXiv, Wikipedia, and Github). Data sampling proportions are a key factor in model performance, with many widely-used LLMs having seemingly arbitrary mixture proportions (Touvron et al., 2023; Team, 2023). Since it is expensive to use trial-and-error to explore the search space of mixture proportions, some recent works have proposed algorithms for data mixing, where proportions are learned based on model performance throughout training (Xie et al., 2023; Albalak et al., 2023). Data mixing scaling laws have also been proposed (Ye et al., 2024). These approaches primarily focus on the pre-training setting and only hold when the training groups of data are the same as the evaluation groups of data, which is not true in our setting of instruction datasets. Skill-It (Chen et al., 2023) proposes an online algorithm for setting proportions when the training tasks are not equal to the evaluation tasks; we note that COOKBOOK-MIX can be considered a non-online instantiation of their approach.

# D    COOKBOOK-MIX additional details

First, we describe the full algorithm for mixing data from templates, with or without labeled evaluation data. Then, we demonstrate that the linear assumption on accuracies empirically holds. Finally, we provide a proof of Proposition 1, the optimal expression for $p^\star$.

## D.1    COOKBOOK-MIX Template Mixing Algorithm

Algorithm 1 presents our approach for how to compute template data proportions. It computes $\hat{p}$ according to Proposition 1; namely, $l$ COOKBOOK-tuned models—one per

---

**Algorithm 1** COOKBOOK-MIX.

---

1: **Input:** $l$ templates $G_T$, base model $f$, $m$ downstream task datasets $\mathcal{D}^{\text{eval}} = \mathcal{D}^{\text{eval}}_1, \ldots, \mathcal{D}^{\text{eval}}_m$, $n$ number of training samples, $\eta$ entropy parameter
2: Fine-tune $f$ on $n$ samples generated from $G_{T_i}$ to get COOKBOOK-tuned model $f_{G_{T_i},n}$ for all templates $G_{T_i} \in G_T$.
3: **if** $\mathcal{D}^{\text{eval}}$ has ground-truth outputs **then**
4:    Evaluate each model $f_{G_{T_i},n}$ on $\mathcal{D}^{\text{eval}}$ to get $\hat{A}_{ij} = \text{acc}(f_{G_{T_i},n}, T^{\text{eval}}_j)$ for all $j \in [m], i \in [l]$.
5: **else**
6:    Set $\hat{A}_{ij} = \text{ESTIMATEACCS}(f_{G_{T_1},n}, \ldots, f_{G_{T_l},n}, \mathcal{D}^{\text{eval}}_j)$, a weak supervision-based method for estimating accuracy without ground-truth outputs, for all $j \in [m], i \in [l]$.
7: **end if**
8: Compute $\sigma_i = \exp(\frac{1}{m\eta} \sum_{j=1}^m \hat{A}_{ij})$ for all $i \in [l]$.
9: Calculate sampling proportion vector $\hat{p}$, where $\hat{p}_i = \frac{\sigma_i}{\sum_{k=1}^l \sigma_k}$ for all $i \in [l]$.
10: **return** $\hat{p}$. $f$ is then fine-tuned on $n$ samples from templates $f_{G_T}$ with proportions $\hat{p}$.

---

**Algorithm 2** ESTIMATEACCS.

---

1: **Input:** $l$ COOKBOOK-tuned models ("voters") $\lambda_i := f_{G_{T_i},n} : \mathcal{X} \to \mathcal{Y}$ for $i \in [l]$, dataset without ground-truth outputs, unlabeled evaluation dataset $\mathcal{D} = \{x^j\}_{j=1}^n$.
2: Get predictions $\{\lambda_1(x^j), \ldots \lambda_l(x^j)\}$ across COOKBOOK-tuned models for each $x_j \in \mathcal{D}, i \in [l]$, forming votes dataset $D_\lambda \in \mathcal{Y}^{n \times l}$.
3: Estimate accuracy $\hat{a}_i \approx \Pr(\lambda_i(x) = y)$ for all $i \in [l]$ by using Algorithm 1 from Ratner et al. (2019) on noisy votes $D_\lambda$.
4: **return** $\{\hat{a}_1, \ldots, \hat{a}_l\}$.

---

template $G_{T_i}$—are trained, and then evaluated on all $m$ downstream tasks. Then, the average accuracy across tasks for each COOKBOOK-tuned model is computed, and a softmax (with temperature $1/\eta$ depending on the entropy regularization in (1)) is performed over these average accuracies to get $\hat{p}$.

When evaluating each COOKBOOK-tuned model, computing accuracy on each downstream task is straightforward if ground-truth outputs are available for the task. However, if they are unavailable but the outputs are discrete (e.g., answer choices in PIQA or "yes" or "no" in ITUNES-AMAZON), we can use techniques from weak supervision to estimate accuracies (line 6 of Algorithm 1). Weak supervision involves producing programmatically labeled data by modeling several weaker "voters" using a latent variable probabilistic graphical model over their votes. In particular, weak supervision algorithms fit the latent variable model using predictions from the voters, estimating the voter accuracies. Then, they aggregate the votes based on the estimated accuracies to get a programmatic label. We utilize a popular weak supervision algorithm from Ratner et al. (2019) to get an estimated accuracy for each COOKBOOK-tuned model in Algorithm 2. In particular, we compute a votes dataset $\mathcal{D}_\lambda \in \mathcal{Y}^{m \times l}$ by using each COOKBOOK-tuned model to produce predictions on the downstream dataset. This votes dataset is converted into numbers (e.g., "yes" becomes 1 and "no" becomes $-1$) and is passed in as input to Algorithm 1 of Ratner et al. (2019). This algorithm uses a conditional independence assumption, which implies a particular sparsity pattern in the inverse covariance matrix of $\mathcal{D}_\lambda$, to create a system of equations that is solved with SGD to yield estimated accuracies.

## D.2 Assessing the Linearity Assumption Empirically

To see if the linearity assumption applies on real data, we consider two templates $G_T = \{G_{T_1}, G_{T_2}\}$, and compare the performance of individual COOKBOOK-tuned models,

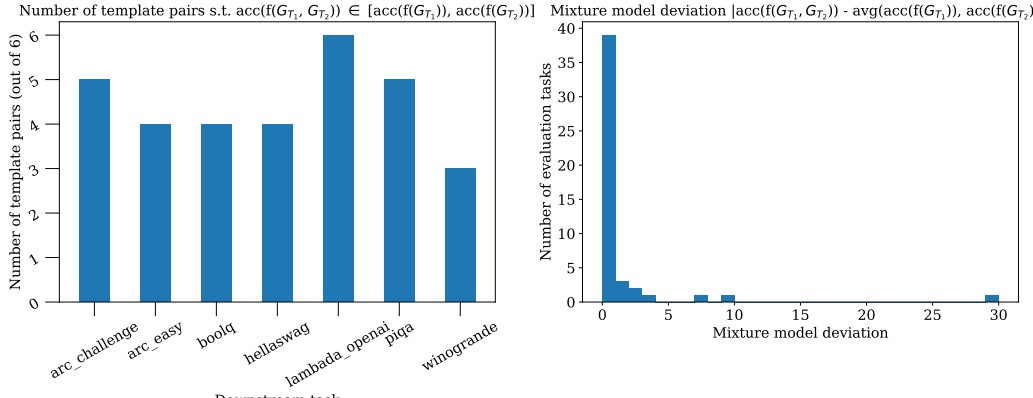

Figure 3: Evaluating if the linearity assumption for data proportions holds empirically. Left: we measure the interpolation property, how often the mixture model has an accuracy in between the individual COOKBOOK-tuned models' accuracies. Right: we measure the mixture model deviation, the absolute difference between the mixture model's accuracy and the average of the individual COOKBOOK-tuned models' accuracies. Measurements are made across 6 pairs of templates (over the 4 templates used in Section 4.1), 8 GPT4ALL evaluation tasks, and the MISTRAL-7B base model.

$f_{G_{T_1},n}$ and $f_{G_{T_2},n}$, versus a model fine-tuned on a uniform mixture over the two templates, $f_{G_T,n[1/2,1/2]}$, which we call the mixture model. We re-use the setting described in Section 4.1 (4 templates, 8 tasks from GPT4ALL, fine-tuning the MISTRAL-7B model). We measure two quantities, described below.

- **Interpolation property:** first, we examine how often the mixed model's accuracy interpolates between the individual COOKBOOK-tuned model accuracies, $\mathrm{acc}(f_{G_T,n[1/2,1/2]}, T_j^{\mathrm{eval}}) \in [\mathrm{acc}(f_{G_{T_1},n}, T_j^{\mathrm{eval}}), \mathrm{acc}(f_{G_{T_2},n}, T_j^{\mathrm{eval}})]$ on all downstream tasks $T_j^{\mathrm{eval}} \in \boldsymbol{T}^{\mathrm{eval}}$. We compute how many pairs of $G_{T_1}, G_{T_2}$ (out of 6 pairs over the 4 templates) for which this condition holds per evaluation task. Our results are shown in Figure 3 (left), where we find that all downstream tasks have at least half of the template pairs satisfying this interpolation property.

- **Mixture model deviation:** second, we measure the absolute difference between the mixture model's accuracy on a task and the average of the individual COOKBOOK-tuned models' accuracies on the task, $|\mathrm{acc}(f_{G_T,n[1/2,1/2]}, T_j^{\mathrm{eval}}) - \mathrm{avg}(\mathrm{acc}(f_{G_{T_1},n}, T_j^{\mathrm{eval}}), \mathrm{acc}(f_{G_{T_2},n}, T_j^{\mathrm{eval}}))|$. We call this the mixture model deviation, and measure this across template pairs and downstream tasks in Figure 3 (right). We find that most values of this deviation are between 0 and 1, meaning that the mixture model's accuracy is less than 1 accuracy point away from the average of individual accuracies.

Based on these results, we do see that there exist some template pairs for which training a model on them results in significantly different performance. This suggests that modeling higher-order interactions among data samples from different templates could help us improve our estimate of $p^\star$, although doing so may result in an optimization problem that lacks a simple closed-form solution.

### D.3 Proof of Proposition 1

We restate Proposition 1 here for completeness.

**Proposition 1.** *Define $A \in \mathbb{R}^{l \times m}$ where $A_{ij} = \mathrm{acc}(f_{G_{T_i},n}, T_j^{\mathrm{eval}})$. Let $\sigma_i = \exp(\frac{1}{m\eta} \sum_{j=1}^{m} A_{ij})$ for all $i \in [l]$. Then, the $\boldsymbol{p}^\star$ that maximizes (1) satisfies $p_i^\star = \frac{\sigma_i}{\sum_{k=1}^{l} \sigma_k}$ for all $i \in [l]$.*

*Proof.* Recall that the objective we aim to maximize is the expression

$$\frac{1}{m} \sum_{j=1}^{m} \sum_{i=1}^{l} p_i A_{ij} + \eta H(\boldsymbol{p}). \tag{3}$$

where $A_{ij} = \mathrm{acc}(f_{G_{T_i},n}, T_j^{\mathrm{eval}})$ and $\boldsymbol{p} \in \triangle^l$, e.g. $\sum_{i=1}^{l} p_i = 1$. The Lagrangian function is thus

$$\mathcal{L}(p, \lambda) = \frac{1}{m} \sum_{j=1}^{m} \sum_{i=1}^{l} p_i A_{ij} + \eta H(\boldsymbol{p}) + \lambda \Big( 1 - \sum_{i=1}^{l} p_i \Big). \tag{4}$$

Taking the partial derivative of the Lagrangian with respect to $p_i$ for some $i \in [l]$ and setting equal to 0, we get

$$\frac{\partial \mathcal{L}(\boldsymbol{p}, \lambda)}{\partial p_i} = \frac{1}{m} \sum_{j=1}^{m} A_{ij} + \eta(-\log p_i - 1) - \lambda = 0. \tag{5}$$

Rearranging, we get

$$\log p_i = \frac{1}{m\eta} \sum_{j=1}^{m} A_{ij} - \frac{\lambda}{\eta} - 1 \tag{6}$$

$$\Rightarrow p_i = \exp\Big( \frac{1}{m\eta} \sum_{j=1}^{m} A_{ij} - \frac{\lambda}{\eta} - 1 \Big). \tag{7}$$

We now plug this expression for $p_i$ into the equation $\sum_{i=1}^{l} p_i = 1$:

$$\sum_{i=1}^{l} \exp\Big( \frac{1}{m\eta} \sum_{j=1}^{m} A_{ij} - \frac{\lambda}{\eta} - 1 \Big) = 1 \tag{8}$$

$$\Rightarrow \exp\Big( \frac{\lambda}{\eta} \Big) = \sum_{i=1}^{l} \exp\Big( \frac{1}{m\eta} \sum_{j=1}^{m} A_{ij} - 1 \Big). \tag{9}$$

Finally, substituting $\exp(\lambda/\eta)$ in the expression for $p_i$ gives us

$$p_i = \frac{\exp\Big( \frac{1}{m\eta} \sum_{j=1}^{m} A_{ij} - 1 \Big)}{\sum_{i=1}^{l} \exp\Big( \frac{1}{m\eta} \sum_{j=1}^{m} A_{ij} - 1 \Big)} = \frac{\exp\Big( \frac{1}{m\eta} \sum_{j=1}^{m} A_{ij} \Big)}{\sum_{i=1}^{l} \exp\Big( \frac{1}{m\eta} \sum_{j=1}^{m} A_{ij} \Big)}. \tag{10}$$

$\square$

# E  Automated Template Creation

In this section, we outline our methodology for auto-generating COOKBOOK templates using GPT-4.

## E.1  Template Generation

We generate task-specific templates by prompting GPT-4. See below for the prompt passed as input to the GPT-4 model. Notice that the prompt has two main components (1) instructions on how to create COOKBOOK templates and (2) two in-context examples mapping a task description to a data generating function.

```
You are an analyst whose job is to help write data generating functions.
```

```
Here is the procedure for writing a data generating template:
* constructing the inputs: inputs are categorized into a
parent input and child inputs; the parent input is a randomly sampled
    ↪ sequence of tokens
from the vocabulary
 * constructing the outputs y\_hat based on the inputs: output
is constructed from the parent and child inputs using a function that
    ↪ approximates the task
rule

Here is a sample data generating template for the task: document question
    ↪  answering

Description of task: Given a document and a questions, retrieve
    ↪ information from the document that answers the question.

Inputs:
--- `Document`: random list of tokens sampled from vocabulary
--- `Question`: randomly selected span from document
Outputs:
--- `Answer`:  span including all tokens +/-3 tokens before the start and
    ↪  end of question span

Data Generating Template:

def qa_template(min_slen: int, max_slen: int):
  document = random.sample(token_ids) #input source

  # inputs: use document to get question
  span_length = random.choice(min_slen, max_slen)
  question = document.get_span(k=span_length) # get indexes

  # outputs: use document and question to get the answer
  answer = document[loc(document.intersect(question)) - 3 : loc(document.
    ↪ intersect(question)) + 3]

    instruction = "Use the document to answer the question.\n"

    return f"""{instruction}
  Document: {document}

  Question: {question}
Answer: {answer}

#####

Here is a sample data generating template for the task: multiple choice
    ↪ question answering

Description of Task: Given a question and a set of 5 answer choices,
    ↪ select the answer choice which best answers the questions

Inputs:
--- `Question`: randomly selected span from vocabulary
--- `Choices`: 5 randomly sampled sets of 5 tokens, where one answer (the
    ↪  correct answer) shares tokens with the question.
Outputs:
--- `Answer`:  answer choice with the maximum overlap with the question.

Data Generating Template:

def multi_choice_qa (overlap_len: int):
  question = random.sample(token_ids)
    (c1, c2, c3, c4, c5) = random.sample(token_ids, k=5)
```

```
  # inputs: use question to get correct choice c5
    c5 = question.sample(k=overlap_len) + c5[:-overlap_len]

  # outputs: use  question, [c1, . . ., c5]) to get the answer
    choices = [c1, c2, c3, c4, c5].shuffle()
  ans_idx = argmax([question \cap choices[0],..., question \cap choices
    ↪ [4]])
  answer = choices[ans_idx]

  instruction = "Answer the question.\n"
  return f"""
  {instruction}
  Question: {question}
  Choices:
  - {choices[0]}
  - {choices[1]}
  - {choices[2]}
  - {choices[3]}
  - {choices[4]}
  Answer: {answer}

####

Write a data generating template for the task: {task_name}

Description of task: {task_description}

Inputs:"""
```

**Auto-generating Templates with External Artifacts** We find that our automated template generation approach is capable of generating external artifacts (such as the rhyming dictionary in Section B.1). See more details here.

### E.2 Evaluation

Using our automated template generation approach, we generate templates for two tasks, entity matching and token retrieval. We observe that for entity matching, the generated template exactly matches—in terms of functionality—the hand generated template (in Appendix B.1). For token retrieval, we observe that the generated template is minorly different than the manually generated template (in Appendix B.6) in that the "question" is sampled as a contiguous span (in oppose to a set of randomly sample tokens) from one of the support documents. A GPT-NEO-1.3B model finetuned on data generated by this template (COOKBOOK-NEO-AUTO), has an accuracy of 18.4: 5.8 points better than the base model and within 0.02 points of a model finetuned on data from the manually generated template (see table below where COOKBOOK-NEO is the performance on manually generated templates).

| Dataset | COOKBOOK-NEO-AUTO | COOKBOOK-NEO |
|---|---|---|
| MS_MARCO | $18.4 \pm 0.01$ | $18.6 \pm 0.16$ |
| BEER | $66.6 \pm 0.0$ | $66.6 \pm 0.0$ |
| ITUNES-AMAZON | $69.6 \pm 0.0$ | $69.6 \pm 0.0$ |

## F Poetry Generation

In this section, we show that the COOKBOOK framework can be extended beyond the three task families to more creative tasks such as poetry generation.

**Task.** We create a poetry generation NL task wherein the model is prompted to write a poem with a prespecified rhyme structure, say ABAB. We construct 30 topics covering a wide range of subjects (e.g., shoe or cup) that the models are then prompted to generate a

poem about. We measure accuracy as how well the model is able to follow the specified rhyme pattern when generating the poem.

**Our method.** We fine-tune the MISTRAL-7B model using a template designed for this poetry task, which we call COOKBOOK-POEM; see Appendix B.7 for the exact template and examples of datapoints. We fine-tune the model for 500 steps with a batch size of 64 where each batch uses fresh samples from the templates. Our template uses a dictionary of rhyme words, but does not use any examples of complete poems.

**Results.** When prompted to generate a 4 line poem with an ABAB rhyme scheme (given 2 in-context examples), base MISTRAL-7B has 20% accuracy, COOKBOOK-POEM has 60% accuracy and GPT-4 has 50% accuracy.

### F.1 COOKBOOK-POEM samples

We present sample poems generated by COOKBOOK-POEM below.

---

**Topic: flower**

```
A flower's petals, so delicate and fair
Its scent is sweet, a perfume so divine
Its beauty's rare, a sight to behold anywhere
Its petals fall, a sign of the decline.
```

---

**Topic: bird**

```
A bird that soars above the clouds so high
Its wings are strong, its flight is free
It's not afraid to try
And soars above the sea
```

---

**Topic: deer**

```
A deer, a female deer, a lovely doe
Its antlers are a symbol of its pride
Amidst the trees, it knows just where to go
To find the food that's on the other side.
```

---

**Topic: galaxy**

```
A galaxy of stars, so far away
Their light, a beacon, in the night
A cosmic dance, a celestial display
A wondrous sight, a sight so bright.
```

---

## G Experimental evaluations: additional details

We provide additional experimental details for both the multi-task evaluation and single-task evaluation settings.

### G.1 Hyperparameter tuning details

For finetuning our COOKBOOK-tuned models, we ran a hyper-parameter searches, sweeping across learning rate ($[4e-06, 5e-06, 8e-06, 5e-05, 8e-05]$), batch size ($[8, 16, 32, 64]$) and total training steps ($[100, 200, 300, 400$ and $500]$).

## G.2 Multi-task evaluation details

### G.2.1 Full GPT4ALL evaluations w/standard deviations

| Model | arc_c | arc_e | boolq | hellaswag | lambada | openbookqa | piqa | winogrande | average |
|---|---|---|---|---|---|---|---|---|---|
| LLAMA-2-7B | 46.25 ± 1.46 | 74.58 ± 0.89 | 77.74 ± 0.73 | 75.99 ± 0.43 | 73.92 ± 0.61 | 44.20 ± 2.22 | 79.11 ± 0.95 | 69.14 ± 1.30 | 67.61 |
| LLAMA-2-7B-flan | 40.61 ± 1.44 | 73.40 ± 0.91 | 77.46 ± 0.73 | 56.87 ± 0.49 | 73.76 ± 0.61 | 31.40 ± 2.08 | 78.62 ± 0.96 | 67.72 ± 1.31 | 62.48 |
| LLAMA-2-7B-self-inst | 40.44 ± 1.43 | 73.27 ± 0.91 | 74.46 ± 0.76 | 57.10 ± 0.49 | 73.37 ± 0.62 | 32.40 ± 2.10 | 78.13 ± 0.96 | 68.82 ± 1.30 | 62.25 |
| LLAMA-2-7B-chat | 44.20 ± 1.45 | 69.74 ± 0.94 | 79.76 ± 0.70 | 75.50 ± 0.43 | 71.08 ± 0.63 | 43.80 ± 2.22 | 77.20 ± 0.98 | 66.46 ± 1.33 | 65.97 |
| LLAMA-2-7B-NH | 49.74 ± 1.44 | 76.09 ± 0.88 | 80.00 ± 0.70 | 77.72 ± 0.42 | 72.99 ± 0.62 | **46.40 ± 2.23** | 79.76 ± 0.94 | 70.01 ± 1.29 | 69.09 |
| MISTRAL-7B | 54.10 ± 1.46 | 79.50 ± 0.83 | 83.49 ± 0.65 | 81.12 ± 0.39 | 75.59 ± 0.60 | 44.40 ± 2.22 | 82.05 ± 0.90 | 73.88 ± 1.23 | 71.76 |
| MISTRAL-7B-inst | 54.18 ± 1.46 | 81.36 ± 0.80 | 85.44 ± 0.62 | 66.04 ± 0.47 | 71.32 ± 0.63 | 35.40 ± 2.14 | 80.30 ± 0.93 | 74.11 ± 1.23 | 68.52 |
| MISTRAL-7B-cap | 54.01 ± 1.46 | 78.54 ± 0.84 | 82.57 ± 0.66 | 78.74 ± 0.41 | 72.46 ± 0.62 | 44.80 ± 2.23 | 79.60 ± 0.94 | 71.03 ± 1.27 | 70.22 |
| MISTRAL-7B-orca | 56.14 ± 1.45 | 79.59 ± 0.83 | 86.57 ± 0.60 | 81.73 ± 0.39 | 72.37 ± 0.62 | 45.60 ± 2.23 | **83.03 ± 0.88** | 73.24 ± 1.24 | 72.28 |
| MISTRAL-7B-OH | **59.98 ± 1.43** | 81.65 ± 0.79 | **86.73 ± 0.59** | **81.77 ± 0.39** | 73.90 ± 0.61 | 44.20 ± 2.22 | 82.70 ± 0.88 | 73.56 ± 1.24 | 73.06 |
| CB-LLAMA | 48.04 ± 1.46 | 76.77 ± 0.87 | 79.20 ± 0.71 | 76.04 ± 0.43 | 77.10 ± 0.59 | 43.40 ± 2.22 | 78.56 ± 0.96 | 69.30 ± 1.30 | 68.55 |
| CB-MISTRAL-UNI | 58.70 ± 1.44 | 82.66 ± 0.78 | 79.97 ± 0.70 | 81.09 ± 0.39 | **78.46 ± 0.57** | 43.60 ± 2.22 | 81.77 ± 0.90 | **75.06 ± 1.22** | 72.66 |
| CB-MISTRAL +WSL | 57.85 ± 1.44 | 82.37 ± 0.78 | 86.39 ± 0.60 | 81.36 ± 0.39 | 77.94 ± 0.58 | 44.60 ± 2.23 | 82.32 ± 0.89 | 74.19 ± 1.23 | 73.38 |
| CB-MISTRAL +WS | 57.76 ± 1.44 | **83.21 ± 0.77** | 85.23 ± 0.62 | 80.99 ± 0.39 | 78.23 ± 0.57 | 44.00 ± 2.22 | 82.32 ± 0.89 | 74.27 ± 1.23 | 73.25 |

Table 5: **Model performance on the GPT4ALL benchmark.** "CB*" denotes our COOKBOOK tuned models where "wsl" is aggregation with labeled evaluation data, "ws" is aggregation without using labels (our COOKBOOK-MIX algorithm) and "uni" is a uniform mixture, "NH" denotes NousHermes Dataset, "cap" is the Nous-Capybara model, "OH" is Open-Hermes, and "orca" is OpenOrca. Averaged across tasks, CB-MISTRAL +WSL and CB-MISTRAL +WS are the best performing models.

### G.2.2 GPT4ALL Benchmark

GPT4ALL (NomicAI) is a standard evaluation benchmark which covers 7 tasks: ARC-Easy (Clark et al., 2018), ARC-Challenge (Clark et al., 2018), PIQA (Bisk et al., 2020), Winogrande (ai2, 2019), BoolQ (Clark et al., 2019), Lambada OpenAI (Radford et al., 2019), OpenBookQA (Mihaylov et al., 2018) and HellaSwag (Zellers et al., 2019).

## G.3 Template mixing evaluation details

| Model | arc_c | arc_e | boolq | hellaswag | lambada | openbookqa | piqa | winogrande | average |
|---|---|---|---|---|---|---|---|---|---|
| CB-MCQA | 55.46 ± 1.45 | 80.43 ± 0.81 | 82.54 ± 0.66 | 80.88 ± 0.39 | 79.27 ± 0.56 | 44.20 ± 2.22 | 81.83 ± 0.90 | 74.35 ± 1.23 | 72.37 |
| CB-MATCH | 57.25 ± 1.45 | 82.37 ± 0.78 | 64.80 ± 0.84 | 81.32 ± 0.39 | 74.60 ± 0.61 | 44.20 ± 2.22 | 82.10 ± 0.89 | 74.19 ± 1.23 | 70.10 |
| CB-ED | 55.46 ± 1.45 | 79.76 ± 0.82 | 82.42 ± 0.67 | 80.59 ± 0.39 | 78.63 ± 0.57 | 44.20 ± 2.22 | 81.56 ± 0.90 | 73.80 ± 1.24 | 72.05 |
| CB-SELECT | 56.14 ± 1.45 | 79.92 ± 0.82 | 83.64 ± 0.65 | 81.10 ± 0.39 | 79.06 ± 0.57 | 43.80 ± 2.22 | 81.50 ± 0.91 | 73.72 ± 1.24 | 72.36 |
| CB-UNI | 58.70 ± 1.44 | 82.66 ± 0.78 | 79.97 ± 0.70 | 81.09 ± 0.39 | 78.46 ± 0.57 | 43.60 ± 2.22 | 81.77 ± 0.90 | 75.06 ± 1.22 | 72.66 |
| CB-WSL | 57.85 ± 1.44 | 82.37 ± 0.78 | 86.39 ± 0.60 | 81.36 ± 0.39 | 77.94 ± 0.58 | 44.60 ± 2.23 | 82.32 ± 0.89 | 74.19 ± 1.23 | 73.38 |
| CB-WS | 57.76 ± 1.44 | 83.21 ± 0.77 | 85.23 ± 0.62 | 80.99 ± 0.39 | 78.23 ± 0.57 | 44.00 ± 2.22 | 82.32 ± 0.89 | 74.27 ± 1.23 | 73.25 |

Table 6: **WS performance analysis** "CB*" denotes our COOKBOOK tuned models on MISTRAL-7B, where -WSL combines templates using downstream datasets with ground-truth outputs, -WS combines templates without ground-truth outputs and -UNI is the uniform mixture. MATCH, ED, MCQA, and SELECT are abbreviations for MATCHING, ENTITY-DISAMBIGUATION, MULTI-CHOICE-QA, COMMONSENSE-SELECT individual COOKBOOK-tuned models.

Table 6 provides additional experimental results around our approach for mixing data from templates. For COOKBOOK-tuned models that use multiple templates, our main method (COOKBOOK plus data proportions obtained using WS on data without ground-truth outputs) is referred to as CB-WS, (previously COOKBOOK-MIST in Table 1). Furthermore, COOKBOOK-WSL shows the results when we use the ground-truth outputs in our evaluation datasets; while this method has slightly higher average accuracy, CB-WS is able to come close despite not having any output information. COOKBOOK-UNI shows the results on a uniform mixture of template-generated data, which performs worse than both previous methods. Table 6 also contains results of the individual models that are COOKBOOK-tuned on MATCHING, ENTITY-DISAMBIGUATION, MULTI-CHOICE-QA, and COMMONSENSE-SELECT; we find that these models' performance is worse than the models that use data from multiple templates.

For the template mixing approach that does not use ground-truth outputs, we use the MeTaL algorithm from Ratner et al. (2019) on each downstream task over 5 random seeds,

learning rate $1e-4$, and number of iterations equal to 5000 (except for BOOLQ, PIQA, and WINOGRANDE, which use 2000).

### G.4 Single-task evaluation details

#### G.4.1 Dataset statistics

We report dataset statistics for the 7 datasets considered in the single-task evaluations. We use the "winogrande-xl" variant of the WINOGRANDE task and an evaluate on v1.1 version of the MS_MARCO dataset. Dataset statistics are listed below in Table 7.

| Dataset | Train | Validation | Test |
|---|---|---|---|
| PIQA | 16.1K | 1.84K | 3.08K |
| SQUAD | 87.6K | 10.6K | None |
| TYDIQA | 151K | 18.7K | None |
| MS_MARCO | 82.3K | 10K | 9.65K |
| WINOGRANDE | 40.4K | 1.27K | 1.77K |
| BEER | 80 | 91 | 91 |
| iTUNES-AMAZON | 156 | 109 | 109 |

Table 7: Summary of dataset sizes for different tasks.

### G.5 Single-task fine-tuning: template to task mapping

Below, we map the COOKBOOK templates used to fine-tune models for the corresponding NL task.

| Dataset | Template |
|---|---|
| PIQA | COMMONSENSE-SELECT |
| SQUAD | DOCUMENT-QA |
| TYDIQA | DOCUMENT-QA |
| MS_MARCO | TOKEN-RETRIEVAL |
| WINOGRANDE | ENTITY-DISAMBIGUATION |
| BEER | MATCHING |
| iTUNES-AMAZON | MATCHING |

Table 8: Task to template mapping.

#### G.5.1 Evaluation details

For all single-task evaluations found in Table 2 and Table 9, we evaluate on 1K examples from the tests sets, with the exception of PIQA, SQUAD and TYDIQA where we sample from the validations sets because they either don't have test sets (SQUAD, TYDIQA) or the test sets are unlabeled (PIQA). We run our evaluations across three different test sets, generated using three separate random seeds.

#### G.5.2 Single-Task fine-tuning: Additional Results

Single-task finetuning results for the CEREBRAS-GPT-1.3B model can be found in Table 9.

## H Analysis: additional experiments

Below, we outline additional experiments conducted to better understand COOKBOOK.

### H.1 Do rules learned on random token distributions transfer to natural language?

We evaluate transferability of rules learned over random tokens by evaluating the performance of COOKBOOK-tuned MISTRAL-7B models on template generated data over *natural*

| Dataset | CEREBRAS-BASE | CEREBRAS-FEW | COOKBOOK-CEREB |
|---|---|---|---|
| TYDIQA | 9.90 ± 0.48 | 8.20 ± 1.85 | **37.50 ± 1.19** |
| SQUAD | 32.10 ± 0.78 | 34.00 ± 3.40 | **51.30 ± 1.00** |
| PIQA | 0.00 ± 0.00 | 47.50 ± 0.61 | **52.80 ± 0.13** |
| MS_MARCO | 6.60 ± 0.49 | 14.40 ± 1.43 | **18.70 ± 1.09** |
| WINOGRANDE | 0.17 ± 0.13 | 30.60 ± 20.00 | **60.30 ± 0.79** |
| BEER | 18.40 ± 0.00 | 26.60 ± 0.00 | **74.10 ± 0.00** |
| ITUNES-AMAZON | 40.00 ± 0.00 | 39.90 ± 0.28 | **55.40 ± 0.00** |

Table 9: Performance comparison of CEREBRAS-GPT-1.3B. Accuracy is reported for all tasks with the exception of BEER and ITUNES-AMAZON for which F1-score is reported.

*language* data samples. Concretely, for template-generated evaluation data over natural language samples, we do not use random tokens but instead apply the input-output rule to natural language. For example, for PIQA and COMMONSENSE-SELECT, a sample could be "Select the choice which best completes the sentence.\nWhen boiling butter, when it's ready, you can \nChoices: \n- glum fray butter boil ready \n- glum fray here crest soar wig". By measuring COOKBOOK's performance on these tasks, we isolate the random token to NL transfer aspect and eliminate additional noise caused by the gap between imperfect rules to true task reasoning. We evaluate COOKBOOK-tuned MISTRAL-7B models on COMMONSENSE-SELECT, MATCHING, and COMMONSENSE-SELECT template rules applied on PIQA, ITUNES-AMAZON, and TYDIQA data, respectively. Our findings show that the rules learned over the random token distributions do transfer to natural language (see Table 10), improving over base performance by up to 29.8 points.

| | MISTRAL-BASE | COOKBOOK-MISTRAL |
|---|---|---|
| **NL-TEMPLATE-PIQA** | 0.044 | 0.326 |
| **NL-TEMPLATE-ITUNES-AMAZON** | 0.666 | 0.964 |
| **NL-TEMPLATE-TYDIQA** | 0.016 | 0.076 |

Table 10: **Random token template to NL-based template data transfer.** Evaluation of random-token-based COOKBOOK-tuned model on natural language-based template data. Rules learned over random tokens transfer to the NL setting.

### H.1.1 When do random tokens work?

Figure 4 shows the effects of pre-training duration on the efficacy of COOKBOOK. Our results indicate that models that have a better understanding of NL (models that are trained longer) have more performance gains from COOKBOOK.

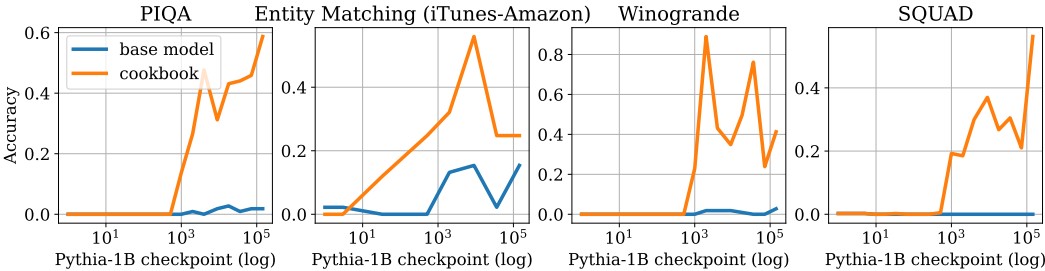

Figure 4: **Effects of pre-training on random token to NL generalization**. Performance gains from COOKBOOK increase with longer pre-training, indicating that maturity of NL understanding is correlated with random-to-NL generalization.

### H.1.2 Do rules taught over random tokens result in less overfitting?

Table 11 compares the degree of overfitting experienced a model finetuned on a natural language task itself, and a model finetuned on a COOKBOOK template. Comparing fine-

tuning on ITUNES-AMAZON versus fine-tuning on MATCHING, our results indicate that rules taught via the template do not hurt base performance on other tasks, but the rule taught by NL-tuning do.

| | MISTRAL-BASE | COOKBOOK-MISTRAL | NL-MISTRAL |
|---|---|---|---|
| ITUNES-AMAZON | 0.696 | 0.823 | 0.875 |
| PIQA | 0.443 | 0.460 | 0.412 |
| TYDIQA | 0.150 | 0.200 | 0.077 |

Table 11: **Rule overfitting.** Comparison of COOKBOOK and NL-tuned models across tasks. The rule taught by template MATCHING, which corresponds to ITUNES-AMAZON, doesn't hurt base performance on other tasks, but the skill taught by NL-tuning does.

### H.1.3 Do we need data generating functions: Are random tokens all we need?

Table 12 shows the results of fine-tuning on a data generated from templates with a format but no rule (i.e., random tokens are used as inputs and outputs without any pattern).

| | MISTRAL-BASE | COOKBOOK-NORULE-MISTRAL |
|---|---|---|
| PIQA | 0.464 | 0.442 |
| ITUNES-AMAZON | 0.697 | 0.195 |
| TYDIQA | 0.178 | 0.151 |

Table 12: **Templates w/o rule.** Evaluation of MISTRAL-7B tuned on data generated from templates with a $\text{fmt}_T$ but no rule.

### H.1.4 Does lots of COOKBOOK-data impair generative ability?

We study the extent to which training on more COOKBOOK data impacts model performance. We run the single-task evaluation from Section 4.2 for 1000 steps (2-10X times more data). We find that there is slight degradation for 3 tasks, while performance on SQUAD improves TYDIQA with more COOKBOOK data.

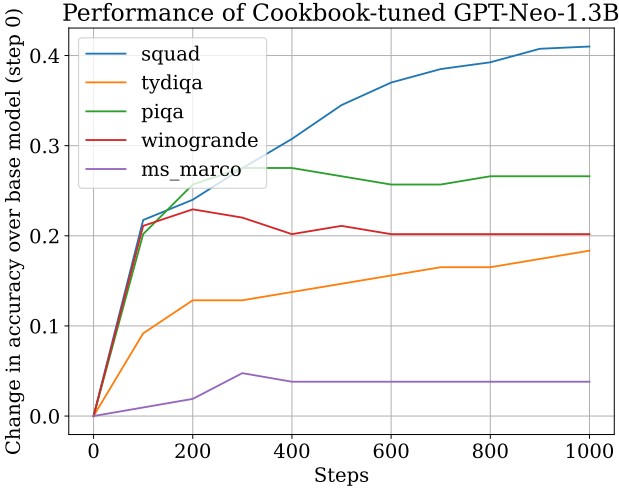

Figure 5: **Effects of training on more COOKBOOK data**. Training on more COOKBOOK data impairs generative ability in some, but not all cases.

