# OpenReview forum: "Cookbook: A framework for improving LLM generative abilities via programmatic data generating templates"
_colmweb.org/COLM/2024/Conference — COLM_

### Official Review · Reviewer_HqBe · 2024-05-09

**Rating:** 6
**Confidence:** 3
**Ethics Flag:** 1

**Summary:**

This paper introduces an automatic framework called Cookbook to construct instruction tuning data without the involvement of human-written data and LLM-generated data. Cookbook generates instruction-tuning data according to programmatic templates of pattern-based rules, and applies the rules to a random token space. To improve performance on multiple tasks, the authors also propose a mixing algorithm over Cookbook templates called Cookbook-Mix. Experiments show that Llama-2 and Mistral finetuned on Cookbook-Mix generated data outperform the base models on the GPT4All benchmark. Analysis shows that the rules taught by templates are crucial to model performance.

**Reasons To Accept:**

1. The idea of constructing instruction datasets via pattern-based rules over random tokens is interesting and courageous. It could be a great starting point and enlighten future works.

2. The paper is well-written. It clearly describes how the templates are constructed and mixed, and how the template alignment scorer works. The analysis section is well-organized and addresses several potential concerns.

**Reasons To Reject:**

1. All the rules designed for tasks in the main content of the paper are based on token overlapping. This may limit the scope of tasks to which this method can be applied, and may teach models to learn shortcuts. For example, the choice that overlaps more with the question is not necessarily the correct answer. The authors acknowledge the restriction and add the task of poetry generation in the appendix, but the template needs more human effort like designing the rhyme_dict, and is unlikely to be generated automatically.

2. The experiments could be further supplemented to make the claims more convincing.

a) The framework could be evaluated on more benchmarks besides GPT4ALL, like Open LLM Leaderboard and BigBench-Hard. More instruction-tuned Llama-2 and Mistral models report their performances on these benchmarks.

b) The instructed versions of Llama-2 and Mistral (Llama-2-chat and Mistral-Instruct) could serve as baselines.

c) For the single-task evaluation, the performance of the 7B models could also be reported.

---

> ### Author Rebuttal · Authors · 2024-05-30
>
> Thank you for your response and your time and effort in reviewing our work. Please find our response to your questions below. We are excited that you find our work interesting and well-written!
>
> 1. Token overlap rules and human effort:
>
> Token overlap is the primary heuristic in our templates, except poetry generation. One may worry that the model learns a shortcut solution from the Cookbook templates, resulting in leveraging overlap-based rules for decision-making. However, our results show that Cookbook leads to performance improvements on downstream tasks—whereas shortcuts are typically concerned with solutions that do not generalize to unseen data. Furthermore, based on our template alignment scorers, we find that the token overlaps (or fuzzy overlaps based on similar words) are indeed present in many natural language samples (see Section 5.1).
>
> For more open-ended tasks that do not rely on token overlap, such as poetry generation, more human effort might be required to generate the templates. That being said, we can use LLMs (i.e., GPT-4) to automate the template generation process (see Section 3.2.2 and Appendix B.7 for more details). In response to your feedback, we also can use LLMs to create the artifacts needed for the template (e.g., the rhyme dictionary in poem generation); using our automated template generation approach, we demonstrate that the model can create its own rhyme_dict in the template below. See template here: https://chatgpt.com/share/32b9e7a5-e69b-4ab7-9eba-0d5e253e2760. We will add this to the final manuscript.
>
> 2. Supplements to Experiments:
>
>    a. Thank you for the suggestion to evaluate Cookbook on OpenLLM and BigBench Hard. We chose to not evaluate on these benchmarks because they require skills not covered by our current template library, such as chain-of-thought reasoning and fact-based QA in various domains, whereas Cookbook focuses on context-based QA.
>
>    b. Regarding adding Llama-2-chat and Mistral-Instruct as baselines, we've included Mistral-Instruct in our GPT4ALL benchmark evaluations. Llama-2-chat was evaluated in the manuscript's appendix (Appendix G.2.1). We find that the best Cookbook model outperforms all these variants. See results here: https://drive.google.com/file/d/1SJpTWHqcOunGQf0eDCHqcUjxM5WWrKur/view?usp=sharing
>
>    c. Great suggestion! We evaluated 1B-scale models to show Cookbook's generalizability across sizes. In the final manuscript, we'll add 7B model evaluations to Section 4.2.

---

> > ### Author Response · Authors · 2024-06-03
> >
> > Dear Reviewer HqBe,
> >
> > We greatly appreciate your valuable feedback and suggestions! To address your concerns we provide empirical evidence as to why the heuristics used in our templates lead to generalized reasoning vs. learning shortcuts, demonstrate how our approach for automated template generation can expand to more complex tasks (i.e., templates for poetry generation that require generating rhyming dictionaries), and expand our experimental baselines to include Llama-2-chat and Mistral-Instruct (which the best Cookbook model outperforms).
> >
> > As the discussion period ends on Thursday, we are happy to answer any additional questions you may have until then. If our response and additional results have addressed your questions and concerns, we would be very grateful if you could acknowledge our rebuttal and consider adjusting your score accordingly

---

> ### Comment · Reviewer_HqBe · 2024-05-31
> **Response to the authors**
>
> Thank you for replying to my concerns and adding the experiments.
>
> However, I may have a different view of shortcuts. Although shortcuts may help models perform well in benchmarks, they are spurious correlations or unwanted biases in the benchmarks. Models that rely on shortcuts may show overestimated capabilities, and could be easily attacked.
>
> [1] Friedman, Dan, Alexander Wettig, and Danqi Chen. "Finding Dataset Shortcuts with Grammar Induction." Proceedings of the 2022 Conference on Empirical Methods in Natural Language Processing. 2022.
>
> [2] Sun, Zechen, et al. "Exploring and Mitigating Shortcut Learning for Generative Large Language Models." Proceedings of the 2024 Joint International Conference on Computational Linguistics, Language Resources and Evaluation (LREC-COLING 2024). 2024.

---

### Official Review · Reviewer_G5a9 · 2024-05-11

**Rating:** 6
**Confidence:** 4
**Ethics Flag:** 1

**Summary:**

The paper introduces Cookbook, a framework designed to generate instruction datasets that enhance large language model (LLM) performance without relying on human or LLM-generated samples. Instead, Cookbook programmatically creates training data using simple patterns over random tokens via Python functions. It generates training data to help the model learn explicit rule-based patterns for specific tasks. The authors also propose a method to mix data from different templates to improve performance across various tasks simultaneously.

**Reasons To Accept:**

**1. Interesting method**: It is very interesting to use code to help us automate the training data generation process. It is a completely novel paradigm than the pure natural language-based methods.

**Reasons To Reject:**

**1. Limited application scenarios**: Based on the examples in Figure 2 and my analysis, the program appears to generate very preliminary span-level or lexical-level training data. It is likely due to the ease of using Python string operations for synthesis. However, this approach seems less effective for generating more semantically complex QA questions or broader user instructions, as Python code alone may struggle to manipulate question semantics effectively. Also, the authors focus their model testing exclusively on QA-related tasks, rather than exploring a wider range of more general tasks that would better demonstrate the framework's broader applicability.

**2. Lack of more in-depth evaluation**: The authors use code to generate training data but overlook other baselines that also facilitate QA data generation. At least, the authors could conduct experiments using Self-Instruct methods to generate QA training data and verify whether the proposed approach truly surpasses Self-Instruct. In my view, particularly for QA tasks, Self-Instruct-style methods appear better suited for generating a broad range of QA questions, irrespective of domain, text length, or format. Moreover, the baselines currently compared to Cookbook weren't specifically trained on QA-style data, potentially contributing to their lower performance relative to Cookbook.

**3. Weird performance linear assumption**: The authors introduce a method to control template proportions in training data based on the assumption that the accuracy of a model trained on a weighted mixture of templates is equivalent to the weighted average accuracy of models individually trained on each template. However, this assumption may be flawed because models specifically trained on one template (e.g., A) tend to perform better on corresponding template-based QA tasks than models trained on a more diverse mixture of templates (e.g., A, B, C, D). In this case, B, C, and D would be out-of-distribution data for the A-based test set. The authors need to conduct experiments to validate this assumption before making claims about its feasibility.

---

> ### Author Rebuttal · Authors · 2024-05-30
>
> Thank you for your response! We appreciate your time and effort in reviewing our work. Please find our response to your concerns below. We are excited that you find our work novel!
>
> 1. Limited Application Scenarios: Cookbook is an initial step in exploring programmatically generated instruction datasets to improve LLM performance across generative tasks. It covers tasks beyond QA, including retrieval, fill-in-the-blank, entity matching, and entity disambiguation (Section 3.2). We acknowledge that token overlap and the list operators we use are not exhaustive for creating templates, as noted in Section 3.2.1. To show Cookbook's broader applicability, we also present a template for Poetry Generation (see Appendix B.7). We consider it future work to extend Cookbook to tasks with more complex instructions beyond the three task families we study (selection, search, and comparison in Section 3.2.1) and their compositions.
>
> 2. Lack of in-depth evaluation: In the original paper, we included results for the best-performing instruction-tuned variants of Llama and Mistral (fine-tuned on NousHermes and Open-Hermes datasets) on GPT4ALL (see Table 1). In response to feedback, we include Mistral-Instruct and Llama Self-Instruct in the table below,  finding that Cookbook-Mistral outperforms both.  Lastly, note that Cookbook-tuned models perform well on GPT4ALL tasks, even those that are not QA tasks and that Cookbook hasn't explicitly learned, such as LAMBADA. See results here: https://drive.google.com/file/d/1SJpTWHqcOunGQf0eDCHqcUjxM5WWrKur/view?usp=sharing
>
> 3. Weird performance linear assumption: We test the validity of the linear data mixing assumption in Appendix D.2, where we take a pair of templates (A, B), and compare the performance of a model trained on A only, B only, and a uniform mix of A and B. We do this over 6 pairs and evaluate on GPT4ALL. We find that the absolute value of the difference between the accuracy of the A-B mix model and the average of the A model and B model is 1.4pts on average, 0.4pts median (Fig 3 right), suggesting that the linear assumption roughly holds.
>
> Also note that this sort of linear assumption is used in recent work on data mixing (https://arxiv.org/abs/2403.16952). Lastly, the assumption is not at odds with your comment about models trained on only A. Our assumption says that the model trained on a mix of ABCD should have performance on A in between one trained on A (good) and one trained on BCD (bad).

---

> > ### Author Response · Authors · 2024-06-03
> >
> > Dear Reviewer G5a9,
> >
> > We greatly appreciate your valuable feedback and suggestions! To address your concerns, we provide clarifications on the broader application of Cookbook (demonstrating its ability to expand to creative tasks such as poetry generation), expand our evaluations to include Mistral-Instruct and Self-Instruct (showing that Cookbook outperforms both of these), and provide empirical evidence as to why our linear assumption holds.
> >
> > As the discussion period ends on Thursday, we are happy to answer any additional questions you may have until then. If our response and additional results have addressed your questions and concerns, we would be very grateful if you could acknowledge our rebuttal and consider adjusting your score accordingly.

---

> > > ### Comment · Reviewer_G5a9 · 2024-06-04
> > >
> > > Thank you for replying to my concerns!
> > >
> > > It addresses part of my questions. I'm more convinced by the strong performance and some empirical validation for the assumptions. Therefore, I will raise my score.

---

### Official Review · Reviewer_cdUR · 2024-05-14

**Rating:** 6
**Confidence:** 3
**Ethics Flag:** 1

**Summary:**

This paper presents an approach, generating instruction dataset prgramatically, in stead of relying on human or LLM based instrcution dataset. Three downstream tasks were selected, where the process of generating datasets were detailed. The paper also indicated that the programmatic data generation contributes to privacy by operating over a random token space, which avoids privacy and legal concerns as it does not involve the use of real user data or violate the terms of service of LLM providers. The evaluation is carried using accuracy  (single and mixed - multitaks setup). The paper is easy to read. However, the way the datasets are selected prgramatically, for example for the QA, which part of the document and what kind of question would be included is not clear. It is mentioned that LLMs are involved in the process but I could not understad the process. Moreover it is not clear or checked how the approach solve the privacy issue.

**Questions To Authors:**

You have a lot of concept in the appendix which makes the paper hard to understand.

**Reasons To Accept:**

- Programatic instruction data generation
- Evaluation approaches in place

**Reasons To Reject:**

- The theoretical appraoch of generating the dataset is sound, but practically, how can we generate (or select ) the dataset from the source is not clear
- The distiniction from the LLM based instruction dataset generation is not clear
- A lot of important content are presented in the Appendixes

---

> ### Author Rebuttal · Authors · 2024-05-30
>
> Thank you for your response! We appreciate your time and effort in reviewing our work. Please find our response to your questions and concerns below.
>
> 1.  How can we generate the dataset from the source: Great question! In Section 3.2.2, we discuss how to generate a template given a task description using generative models such as GPT-4. Using this LLM-generated template, we can construct a dataset from the source. Once we have a collection of templates available, we provide details in Section 3.3 on how to mix data from the templates to perform well on a set of downstream tasks. In the final version of the manuscript, we will expand this discussion.
>
> 2. Distinction between LLM-based instruction datasets: We believe that there are two main distinctions between the template-generated datasets in Cookbook and the LLM-based instruction datasets. The first is the distinction in the data generation process: LLM-based datasets use LLMs (i.e., GPT-4) to generate input/output pairs whereas Cookbook generates input/output data programmatically. As a result, the data generated by Cookbook avoids privacy and legal issues and is more scalable and cost-effective (i.e., using an LLM to generate a sample for a document QA task takes ~21.6 seconds, and costs 1 cent, while using the Cookbook template takes 0.00007 seconds and costs $0). The second is that Cookbook samples attempt to approximate task rules (i.e., reasoning patterns such as localize and retrieve for QA), whereas LLM-based instruction datasets provide natural language demonstrations of a task. This is akin to teaching mathematics by focusing on the underlying principles and problem-solving strategies that can be adapted to various problems, rather than just showing specific solutions to particular problems.
>
> 3. Important content is provided in the appendices: We apologize and will move the method and experimental details from the Appendix to the main paper in our final draft.

---

> > ### Author Response · Authors · 2024-06-03
> >
> > Dear reviewer cdUR,
> >
> > We greatly appreciate your valuable feedback and suggestions! To address your concerns, we have provided clarifications as to how we generate Cookbook templates in an automated manner and how we select templates for downstream tasks. We have also provided more details on the key distinctions between Cookbook and LLM-based datasets, demonstrating empirically that Cookbook is more scalable and cost-effective.
> >
> > As the discussion period ends on Thursday, we are happy to answer any additional questions you may have until then. If our response and additional results have addressed your questions and concerns, we would be very grateful if you could acknowledge our rebuttal and consider adjusting your score accordingly.

---

### Official Review · Reviewer_5LFp · 2024-05-19

**Rating:** 7
**Confidence:** 3
**Ethics Flag:** 1

**Summary:**

This paper introduces COOKBOOK, a new framework designed to boost the generative abilities of LLMs with programmatically generated training data. Unlike recent methods that rely on costly human-curated or LLM-distilled datasets, COOKBOOK uses Python functions to create data templates that could generate random token samples as the instruction-tuning instances. These templates teach models specific pattern-based rules, making the data generation process scalable and free from privacy concerns.

The authors show that models fine-tuned with COOKBOOK could significantly improve their performance on various NLP tasks. Besides, the COOKBOOK-MIX algorithm can select data from multiple templates to further enhance performance. Experimentally, the results are impressive. Models fine-tuned on COOKBOOK beat other instruction-tuned LLMs on several benchmarks. The authors also conduct in-depth analyses to show that why templates based on random token sequences can improve model performance.

**Questions To Authors:**

I have a question regarding data size. Intuitively, if too much unreadable data is introduced during the SFT stage, it might potentially degrade the model's natural language understanding capabilities. Did the authors conduct experiments on this? Although the proposed method can generate unlimited data scalably, could introducing too much of this data gradually impair the model's overall generative ability?

**Reasons To Accept:**

1. Using templates to generate instruction-tuning data with random token sequences is a novel idea for me. It appears that LLMs have sufficient natural language understanding to learn instruction-following abilities even from pattern-based, unreadable pseudo-data.
2. The authors conducted extensive experiments on different LLMs across various benchmarks, and the results are impressive, strongly supporting their claims.
3. Additionally, the authors provided empirical analyses to explain why these template and random token sequence-based data still work effectively.

**Reasons To Reject:**

1. The baselines (instruction-tuning datasets) compared by the authors seem to focus primarily on open-ended conversations, while the tests mainly involve traditional NLP tasks. Including more instruction-tuning datasets on these NLP tasks like FLAN could make the experiments more convincing.
2. The experimental descriptions in the paper are sometimes too brief, making it hard to follow. For instance, the sizes of the different datasets used (both baseline and COOKBOOK) and various training details are either omitted or scattered across the Appendix.
3. I find the design of the QA-related templates unclear and not well-motivated. For example, for Document QA, why is the answer set to the k tokens before and after the question? And why is k set to 3 (Appendix B.3)?

---

> ### Author Rebuttal · Authors · 2024-05-30
>
> Thank you for your response! We appreciate your time and effort in reviewing our work. Please find our response to your questions below. We are glad that you found our work novel and impressive.
>
> 1. Including more instruction-tuned baselines could help: In our paper we include the results for the best performing instruction-tuned variants of Llama and Mistral (namely those fine tuned on the NousHermes and Open-Hermes datasets) on GPT4ALL (see Table 1 in the paper). In response to feedback, we have included other instruction-tuned variants such as Llama-FLAN (https://huggingface.co/allenai/open-instruct-flan-v2-7b), Llama Self-Instruct (https://huggingface.co/allenai/open-instruct-self-instruct-7b), and Mistral-Instruct (https://huggingface.co/mistralai/Mistral-7B-Instruct-v0.2). We find that the best Cookbook model outperforms all these model variants. See results here: https://drive.google.com/file/d/1SJpTWHqcOunGQf0eDCHqcUjxM5WWrKur/view?usp=sharing
>
> 2. Experimental descriptions: Thanks for the feedback. We'll move the dataset size and training details from Appendix G to the main paper in our final draft!
>
> 3. Motivation behind design of QA-template: The template includes tokens before and after the answer to simulate real-world QA, where the answer can be before or after the question’s subject (for example, the question “What is [the capital of France]?” could be answered in the passage with “Paris is [the capital of France]” or “[The capital of France] is Paris”). k = 3 in the template is chosen through hyperparameter tuning on the validation set of the QA datasets in Table 2. We'll include these details in the final draft.
>
> 4. Dataset size and impairing generative ability: We study the extent to which training on more Cookbook data impacts model performance. We run the single-task evaluation from section 4.2 for 1000 steps here: https://drive.google.com/file/d/1SOfofIPDqbfePqfii6enfPxw2Ww9-a4n/view?usp=sharing, whereas the results in table 2 are based on a sweep of 100-500 steps. There is slight degradation for 3 tasks, while performance on SQuAD and TyDi QA improves with more Cookbook data.
>
> We also find that mixing small amounts of natural language samples with Cookbook data preserves NL abilities while teaching template skills. Training on a mix of 128 NL samples and Cookbook data outperforms training on NL samples alone here: https://drive.google.com/file/d/1HHMHkzTEmCFsHZgXjtypGl0YIuLV0zjD/view?usp=share_link

---

> > ### Author Response · Authors · 2024-06-03
> >
> > Dear Reviewer 5LFp,
> >
> > We appreciate your valuable feedback and suggestions. Thank you for raising your score. Please let us know if you have additional questions!

---

### Decision · Program_Chairs · 2024-07-10

**Decision:**

Accept

**Comment:**

The authors consider a framework of generating training data for LLMs by using Python functions, showing that the resulting LLMs trained by such instances outperform their counterparts using other data such as standard SFT data and LLM generated data.

The idea is interesting. The main criticism comes from the fact that the three tasks listed in the main submission rely on certain lexical patterns, which might not represent a wide variety of downstream tasks. The authors discussed this fact int he rebuttal, referring to a very different task in the supplementary material. Some reviewers questioned the selection of LLMs for testing, and the author presented additional results. Overall, the rebuttal was convincing to most reviewers.